



# Seismo-acoustic and GNSS monitoring of a record-breaking storm in the Black Sea: Evidence of climate change and intensifying natural hazards

Laura Petrescu[1,2,*], Bogdan Antonescu[1,2], Sorin Nistor[3], Iustin Floroiu[4,5], Dragoș Ene[1], Daniela Ghica[1], Constantin Ionescu[1], Andrei Anghel[4], Mihai Datcu[4]

1. National Institute for Earth Physics, Magurele, Romania
2. University of Bucharest, Faculty of Physics, Magurele, Romania
3. University of Oradea, Faculty of Construction, Cadaster and Architecture, Oradea, Romania
4. Politehnica University of Bucharest, Faculty of Electronics, Telecommunications and Information Technology, Bucharest, Romania
5. Politehnica University of Bucharest, Doctoral School of Electronics, Telecommunications & Information Technology, Bucharest, Romania

* laura.petrescu@infp.ro

**Abstract**

In August 2024, a devastating storm struck Romania's Black Sea coast, setting new precipitation records and highlighting the increasing frequency of extreme weather events. This study explores the integration of non-conventional sensors (seismic, GNSS, infrasound, and satellite data) with ERA5 meteorological reanalysis to monitor storm dynamics. High-frequency (>30 Hz) seismic signals captured precipitation, while microseismic bands (0.1-1Hz) reflected wave-induced ground motion. Infrasound data, analyzed using unsupervised learning, revealed distinct storm phases and showed strong spectral correlation with recorded ground motion, pointing to coupled atmosphere-lithosphere processes induced by the storm. The infrasound array also detected over 1,100 signals in the 0.6-7 Hz band, matching lightning discharges observed by geostationary satellites. GNSS-derived estimates of precipitable water vapor tracked atmospheric moisture buildup and showed clear correlation with intense rainfall, including potential precursory signals days before peak precipitation. This study highlights the value of integrating diverse, non-traditional datasets to enhance the resolution and depth of storm analysis. Their combined use offers a more holistic understanding of storm evolution and supports the development of improved early-warning systems in vulnerable coastal regions.




## Graphical abstract


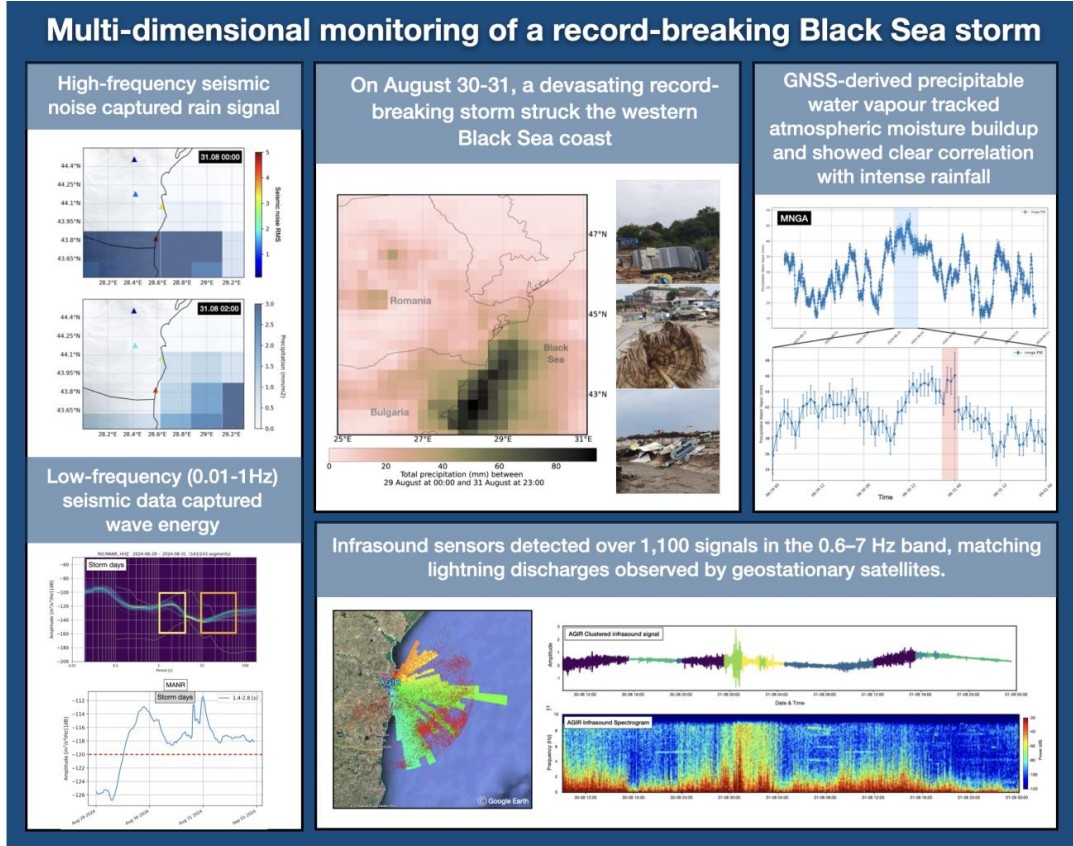











## 1. Introduction

Climate change has become a critical global issue, with far-reaching effects on weather patterns and the frequency and intensity of extreme events (Stott, 2016). These changes are not only contributing to more severe weather events but also altering the timing, location, and duration of storms, making them harder to predict and manage (Bengtsson et al., 2006). Understanding how to effectively monitor and predict the behavior of storms, particularly extreme ones, is crucial for improving forecasting models, enhancing early warning systems, and mitigating their impacts on both natural and human systems.

Traditional meteorological monitoring relies heavily on ground-based stations, weather radars, and satellite observations to track and predict storm behavior (Kober and Tafferner, 2009). These systems have been the backbone of weather forecasting for decades, providing valuable data on temperature, pressure, wind speed, and precipitation. However, while these methods are effective, they often have limitations in terms of spatial coverage (e.g. Sokol et al., 2021), particularly in remote or hard-to-reach areas. Additionally, they may struggle to capture certain atmospheric phenomena in real-time. As a result, non-conventional monitoring methods are increasingly being integrated into storm tracking efforts to complement existing meteorological approaches (e.g. Bosy et al., 2012; Burtin et al., 2016; Diaz et al., 2023; Coviello et al., 2024) .

In this context, our study focuses on the integration of alternative environmental datasets, including GNSS stations, infrasound sensors, and seismic data, to track the dynamics of an extreme storm event, as part of a national climate change resilience strategy, implemented through the DTE Climate project (https://dteclimate.upb.ro/). GNSS data provide valuable information on atmospheric water vapor, helping to track moisture changes that influence storm formation and intensity (Bosy et al, 2012; Marut et al., 2022). Infrasound sensors detect low-frequency acoustic waves generated by storm activity, such as lightning or large-scale weather system movements like microbaroms (e.g. Landès et al., 2012). Seismic data, though traditionally used for earthquake monitoring, can also record vibrations caused by storm-induced pressure changes, making it useful for detecting rainstorms, floods, or tropical cyclones (e.g. Retailleau and Gualtieri, 2021). Through the integration of these diverse sensor networks, our work highlights their synergy in improving storm detection, monitoring capabilities, and early warning systems, contributing to more robust climate resilience strategies.

## 2. Study area and storm overview

The Black Sea region (Figure 1) is characterized by a unique combination of geographic and meteorological features that significantly influence its climate and weather patterns. Nestled between Europe and Asia, the Black Sea is bordered by six countries with diverse landscapes, from mountainous areas to flat plains. This geography, combined with the Black Sea's relatively shallow waters compared to oceanic environments and its connection to the Mediterranean through the Bosphorus Strait, creates an environment where rapid changes in weather are common. Understanding the dynamics of these extreme weather events is crucial, as they can have a profound impact on the environment, economy, and daily life in the region. Monitoring such events is key to improving our ability to predict their occurrence and intensity. By studying the complex atmospheric processes that govern these storms, we can enhance predictive models and refine early warning systems, ultimately helping to mitigate the risks and protect the communities and ecosystems most vulnerable to these extreme weather phenomena.

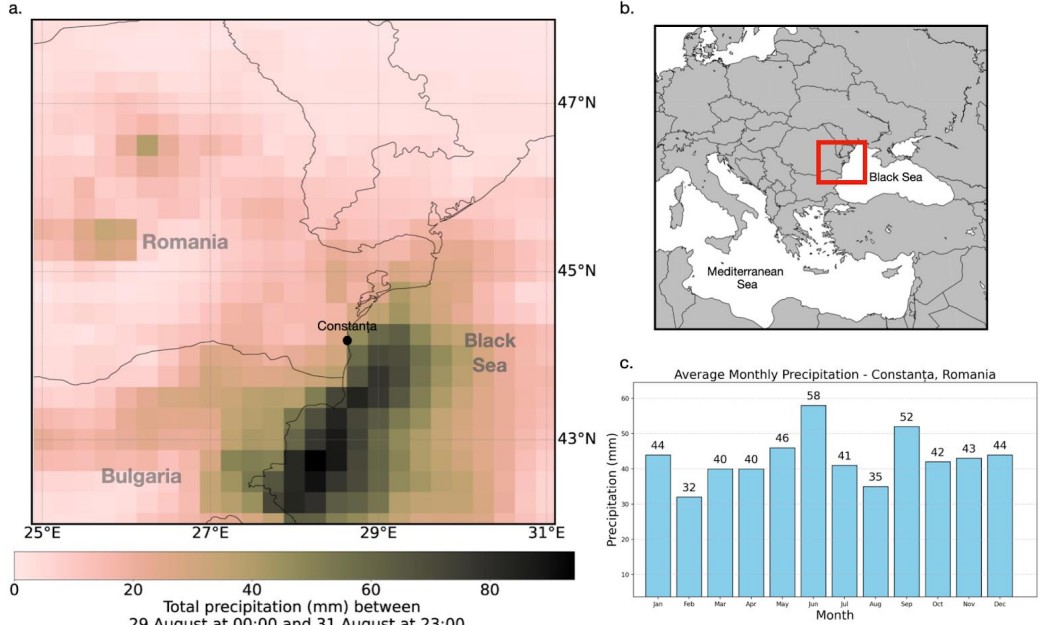

*Figure 1. a. Total precipitation accumulated (in mm, shaded according to the scale) between 29 August 00 UTC and 31 August 23:00 UTC extracted from ERA5 data. The positions of the seismic stations in eastern Romania are also indicated; b. Map of Europe with red square marking the study region; c. Average monthly precipitation rates in Constanța, Romania.*

In August 2024, Romania experienced severe flooding, largely driven by a storm that brought unusual precipitation patterns to the Black Sea coastal region. Exceptional precipitation totals were recorded over south-eastern Romania in Mangalia (225.9 mm), Agigea (145 mm), and Tuzla (118 mm), leading to significant flooding in coastal towns (Figure 1). Over 800 emergency calls prompted large-scale intervention by ISU Dobrogea, focusing on evacuations, debris clearance, and infrastructure restoration (Antonescu et al. 2024).

An analysis conducted by ClimaMeter ([www.climameter.org](www.climameter.org), Faranda et al. 2024, Antonescu et al. 2024) immediately after the event, showed that low pressure systems similar to the one that caused the floods typically result in reduced rainfall (7 mm day$^{-1}$, or up to 15% less) in eastern Romania compared to historical trends. However, this particular storm led to a significant local increase in precipitation, particularly in Constanța, one of the coastal cities severely affected by the flooding. In Constanța, daily rainfall reached up to 5 mm day$^{-1}$, or up to 10% more than usual, marking a notable deviation from the region's typical weather behavior. The changes in precipitation that contributed to the flooding are largely attributed to human-induced climate change, with natural climate variability likely playing a modest role. As climate change continues to influence weather patterns, understanding the connection between changing precipitation levels and extreme weather events like flooding is crucial for improving forecasting and resilience in the face of such disasters.



## 3. Data and Methods

The analysis of the storm event integrates a variety of data sources and methodologies to provide a comprehensive understanding of its dynamics. Seismic data, infrasound measurements, GNSS water vapor data, and ERA5 reanalysis data are all utilized to capture different aspects of the storm's behavior (Figure 2). Seismic data offers insights into ground vibrations and atmospheric disturbances, while infrasound monitoring detects low-frequency acoustic signals generated by lightning and other meteorological phenomena. GNSS water vapor data provides valuable information on atmospheric moisture. Additionally, ERA5 reanalysis data (Hersbach et al. 2020), which provides detailed atmospheric and wave dynamics data, helps contextualize the storm's impact within broader weather patterns. Together, these diverse data sources enable a multifaceted approach to studying the storm and its effects.

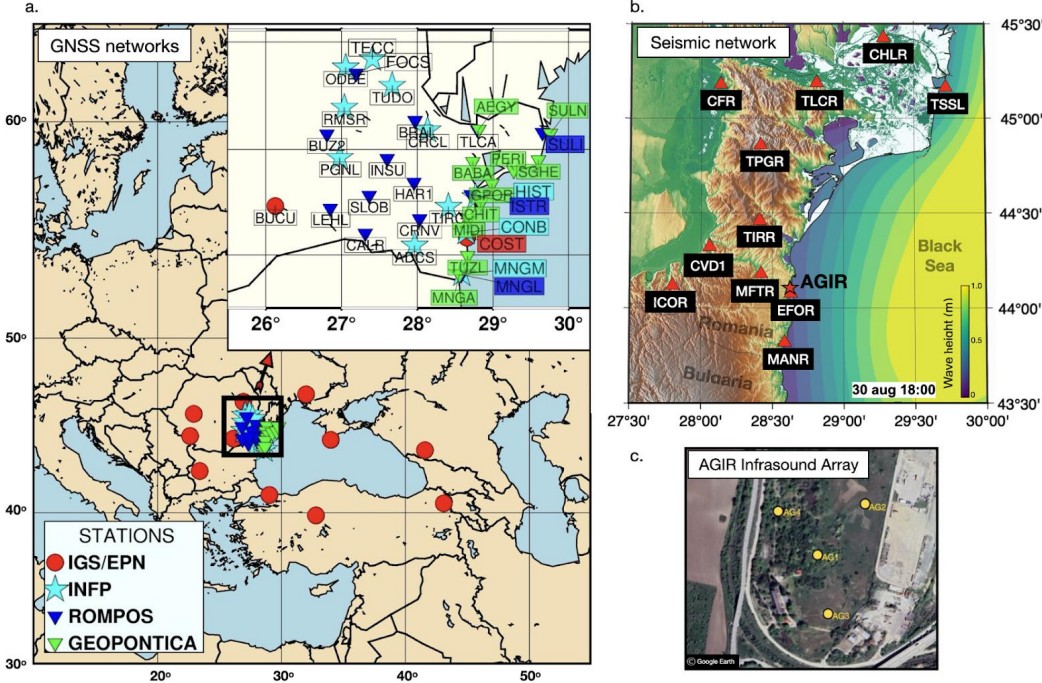

*Figure 2. a. Map of Europe and the Black Sea coast showing GNSS stations and their belonging networks used for analysis in this study. b. Map of the Romanian sea coast showing seismic stations (red triangles) and the location of the AGIR infrasound array (star). Coloured contours represent total wave height at 18:00 on the 30th of August 2024 from ERA5 reanalysis data. c. The layout of the AGIR infrasound array.*



137

## 3.1 Seismic Data

Seismic data represents vibrations of the Earth's surface, commonly referred to as seismic noise. These low-amplitude movements are recorded across the Earth's surface and are traditionally used to study the Earth's internal structure and detect earthquakes. Recently, it has increasingly found applications in meteorology and hydrology, particularly for monitoring weather events (e.g. Dias et al. 2023; Hua et al., 2023), destructive flood episodes (Burtin et al., 2016), ocean storms, and tropical cyclones (Gualtieri et al., 2018). Seismic noise can reveal the impact of atmospheric and oceanic conditions, providing valuable insights into weather events and climate changes (e.g. Bromirski etnal.l 2002; Aster et al, 2008; 2023). In particular, seismic data helps track variations in the Earth's surface caused by factors such as ocean waves, wind, and precipitation, offering a unique perspective on these phenomena (e.g. Grevemeyer et al., 2000; Borzì et al., 2022).

Seismic data also reveals two primary peaks (Figure 3) related to ocean wave interactions (Koper et al., 2015; Ardhuin et al., 2019; Tanimoto et al., 2023). The primary peak, observed in the range of 10–20 seconds (0.05–1 Hz), is generated by the impact of "swell" waves traveling in the same direction, inducing pressure variations in the Earth's crust that match the period of the waves. The secondary peak, in the range of 5–10 seconds (0.1–0.5 Hz), is produced by wind-driven waves, which propagate in different directions and generate pressure oscillations on the ocean floor (Ebeling et al., 2012). These seismic signals directly link ocean conditions with seismic activity (Li et al., 2020), providing insights into large-scale weather phenomena like ocean storms.




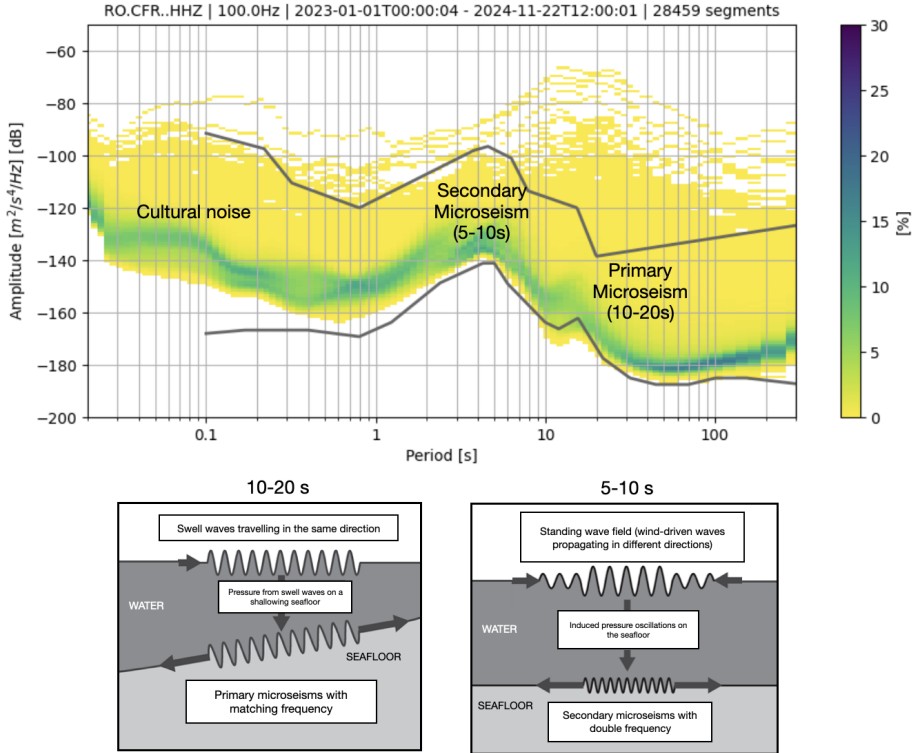

*Figure 3. Probabilistic Power Spectral Density (PPSD) of seismic noise for station CFR, over two years, showing key sources of primary and secondary microseisms. Below, sketches illustrate the generation mechanisms: primary microseisms are caused by unidirectional swell waves inducing pressure fluctuations on a shoaling seafloor, while secondary microseisms result from nonlinear interactions of wind-driven waves over deeper water (modified after Ebeling, 2012).*

Higher frequencies above 30 Hz are associated with the effects of precipitation and wind, as seen in studies like Rindraharisaona et al. (2022) or Diaz et al. (2023). These higher-frequency seismic signals help track more localized weather events, such as storms and heavy rainfall. Seismic data, when integrated with other meteorological tools, enhances the ability to monitor and predict weather events.

To analyse seismic data, the raw traces are first corrected for instrument response and converted to units of velocity. These are then filtered with butterworth filters adapted to capture the target signal: low pass filtering (<1 Hz) for wave-seafloor coupled interactions and high pass filtering (>30 Hz) to identify possible signatures of precipitation, essentially induced pressure fluctuations in the ground converted to weak seismic vibrations due to rain drops. Spectrograms of these filtered seismic traces are also used to visualise the potential signature of the two hydro-meteorological phenomena in the frequency content of ground vibrations.

Potential environmental signals in the seismic data were also investigated using power spectral density (PSD) analysis. To account for variations over time, a Probabilistic Power Spectral Density



(PPSD) method is applied. This approach combines PSD estimates from overlapping time
windows into a probability distribution, providing a comprehensive view of the range and
likelihood of noise levels at different frequencies. Typical noise conditions, as well as transient or
extreme events, are captured in this analysis. To compute the PPSD, ObsPy was used (Beyreuther
et al.,, 2010), which handles data gaps and ensures reliable normalization.
Temporal variations in PSD amplitudes are also analyzed to track changes in seismic noise at
specific frequencies. By extracting PSD values at selected frequencies that are expected to capture
primary and secondary microseisms, time series of noise levels are generated. These temporal
PSDs allow for the identification of trends and correlations with environmental factors, such as
ocean wave activity or weather conditions.

### 3.2 Acoustic Data

Infrasound waves are low-frequency acoustic waves that are inaudible to the human ear, typically
below 20 Hz. These waves are generated by a variety of natural and anthropogenic sources,
including meteorological events, volcanic eruptions, earthquakes, and human activities such as
explosions and industrial processes (Campus, 2009; Bondár et al., 2022). In particular, infrasound
is often associated with phenomena like thunderstorms, ocean waves, and large-scale atmospheric
events, which generate pressure fluctuations that propagate through the atmosphere (e.g. Stopa et
al., 2012; Landès et al., 2012; Listowski et al., 2022). These waves provide valuable information
about the dynamics of weather systems (e.g. Hupe et al., 2019), making them an essential tool for
monitoring and understanding environmental processes (e.g. Brachet et al., 2009; Hupe et al.,
*2022*).
For the monitoring of infrasound signals, we use data from an infrasound array system located at
Eforie Nord-Agigea, Romania (AGIR, Figure 2). This array consists of multiple sensors, including
SIS-1 infrasonic sensors (Seismowave), equipped with global positioning systems (GPS) and noise
reduction technology.
To analyze the seismo-acoustic characteristics of the August 30–31 Black Sea storm, we used a
two-pronged approach: (1) single-station signal analysis based on feature extraction and
unsupervised machine learning, and (2) array-based analysis using classic multi-channel
correlation algorithms. Together, these methods provide complementary insights into the acoustic
behavior of the storm, capturing both local signal characteristics and spatial coherence across
sensors.
For the single-station analysis, infrasound data recorded at the AGIR sensor (Figure 2) was
segmented into 30-minute windows, and a set of time-frequency features was extracted to
characterize the signal dynamics. These features describe how energy and frequency content
evolve over time, providing insights into the signal's structure. The spectral centroid, spectral flux,
spectral rolloff, and spectral entropy capture various aspects of the frequency distribution and its
complexity, while the zero-crossing rate, mean, and variance of the power spectrum reflect signal
activity and variability. These features together form a multidimensional representation of the
infrasound signal during the storm.
The extracted features were used as input for K-Means clustering (MacQueen, 1967), an
unsupervised machine learning algorithm that partitions data into a predefined number of groups,
in this case, seven. K-Means minimizes within-cluster variance by iteratively assigning feature



vectors to the nearest cluster centroid and updating the centroids based on the grouped data. This
clustering method enables the identification of distinct acoustic patterns in the signal (e.g. Pásztor
et al., 2023), offering a data-driven way to segment the storm's infrasound profile without requiring
prior labels or assumptions.
In parallel with the single-station analysis, we also applied the Progressive Multi-Channel
Correlation (PMCC) method (Cansi and Le Pichon, 2008; Le Pichon et al., 2010) to detect and
analyze coherent acoustic signals across an infrasound array. The PMCC method targets signals
generated by atmospheric sources such as lightning or pressure disturbances, operating in the low-
frequency range of 0.7 to 7 Hz. It is specifically suited for mini-array configurations, where signal
coherence between closely spaced sensors can be exploited for precise signal detection and
characterization.
The PMCC algorithm divides waveform recordings into overlapping time windows and processes
them across logarithmically spaced frequency bands. Within each time-frequency segment, cross-
correlations are computed between all sensor pairs to identify coherent wavefronts, signals that
exhibit consistent arrival times across the array. From these detections, PMCC estimates several
key propagation parameters, including back-azimuth (the direction of arrival), horizontal trace
velocity, amplitude, duration, and dominant frequency. This approach is particularly effective in
noisy environments and enables the discrimination of storm-generated infrasound from
background signals or unrelated acoustic sources. The algorithm's output consists of a time-
frequency map of signal detections enriched with physical metadata, allowing for detailed
interpretation of the storm's acoustic footprint and its temporal evolution.

## 3.3 Satellite Observations
We also incorporated data from the Meteosat Third Generation (MTG) satellite system (Holmlund
et al., 2021), specifically from its Lightning Imager (LI) sensor (Viticchie et al., 2020). The MTG
satellites operate in geostationary orbit at approximately 36,000 km altitude, providing continuous
observations over Europe, Africa, and surrounding waters. The Lightning Imager detects cloud-
to-cloud, cloud-to-ground, and intra-cloud lightning flashes using four cameras that collectively
cover 86% of the Earth's visible disc from the satellite's perspective.
For this study, we used Level 2 group data, which includes the geographical coordinates and timing
of each detected flash. By mapping these detections, we were able to analyze the spatial
distribution and temporal evolution of the storm's lightning activity. The dataset also offered
insights into the storm's intensity and structure, complementing other meteorological observations.
## 3.4 GNSS Data
The use of GNSS technology for atmospheric monitoring provides a powerful tool for analyzing
extreme weather events. Beyond its well-known applications in navigation and timing, GNSS has
become a reliable method for sensing tropospheric water vapour, an essential driver of weather
systems and a key variable in forecasting models (Guerova et al., 2016; Vaquero-Martínez and
Antón, 2021). Over the past two decades, ground-based GNSS networks in Europe have
contributed significantly to operational meteorology by providing near real-time estimates of
atmospheric water vapour, aiding in the detection and tracking of severe weather, including heavy



rainfall and storms (Karabatić et al., 2011; Priego et al., 2017; Jones et al., 2020). These high-
resolution observations have proven valuable for both nowcasting and validating numerical
weather prediction models (Wilgan et al., 2015; Bosy et al., 2012; Awange, 2012).
In this study, GNSS data were collected from several networks (Figure 2), including the
International GNSS Service (IGS, Johnston et al., 2017), the EUREF Permanent Network (EPN,
Bruyninx et al,, 2012), the Romanian Position Determination System (ROMPOS, Iliescu et al..,
2019), and GEOPONTICA (Dimitriu et al., 2017). A total of 37 permanent GNSS stations were
analyzed over a 30-day period, with the rainiest interval selected at the midpoint of the study
period. These stations provide high-quality, continuous observations critical for atmospheric
monitoring.
The data were processed using a double-differenced, ionosphere-free combination of L1 and L2
carrier phases. This approach helps minimize errors such as ionospheric delays, satellite clock
biases, and other common atmospheric effects. The resulting Zenith Tropospheric Delay (ZTD)
values were then corrected using the Vienna Mapping Functions 3 (VMF3, Landskron et al., 2018),
which improve the accuracy of ZTD by accounting for variations in the troposphere's atmospheric
conditions. Once the ZTD was refined, it was converted into integrated precipitable water vapor
(PWV) using surface meteorological data (temperature and pressure) from co-located weather
stations, following the method outlined by Bosy et al. (2012). This process allowed for the
derivation of high-resolution atmospheric water vapor content, critical for analyzing the dynamics
of the extreme storm event over the Black Sea. By combining GNSS-derived PWV with data from
other observational sources, the study captured the temporal and spatial variations in atmospheric
moisture, offering valuable insights into the storm's development and intensity.

**3.5 Meteorological Data**
To compare the infrasound signals captured during the Black Sea extreme storm event, we
extracted meteorological data from the open-access ERA5 reanalysis dataset, produced by the
European Centre for Medium-Range Weather Forecasts (ECMWF). This dataset provides a
comprehensive record of global weather conditions from 1950 to the present (Hersbach et al.,
2023). ERA5 combines observational data and advanced numerical models to generate high-
resolution atmospheric parameters, including precipitation (Figure 1), wind speed, and wave
height.
For our study, the ERA5 data was used to track the meteorological context of the storm, offering
insights into the intensity of precipitation, the evolution of wind patterns, and the development of
oceanic wave heights. With high temporal (1 hour) and spatial (0.25° x 0.25°) resolution, ERA5
allows for a detailed comparison of the storm's meteorological characteristics over time. These
comparisons help us understand the storm's dynamics and assess its impact, further enhancing the
interpretation of infrasound signals and aiding in future storm prediction and monitoring efforts.
The open-access nature of ERA5 ensures broad accessibility, contributing to the transparency and
reproducibility of our storm analysis (Copernicus Climate Change Service, Climate Data Store,
299 2023).






## 4. Results

### 4.1 Seismic signatures of storm evolution

High frequency (>30 Hz) analysis of seismic noise reveals strong signals at times when high precipitation values were also recorded (Figure 4). Specifically, an example of the velocity envelope at station MANR and its spectrogram plotted for the period 29th of august at 12 AM (00:00) to 31st of August midnight UTC (Figures 4b,c) reveal strong signal around the 18:00 UTC - 00:00 UTC, when most precipitation was estimated in the grid cell of the station (more than 3 mm m$^{-2}$, Figure 4a). Similar signatures and precipitation-seismic spectrogram visualisations showed correlation with hourly precipitation levels, indicating that the high amplitude signal present on the high frequency velocity envelope and the time-frequency spectrogram at frequencies above 30 Hz is likely caused by rain. Previous research showed a correlation with rain drop size as well (Rindraharisaona et al., 2022), although we did not have access to such information.

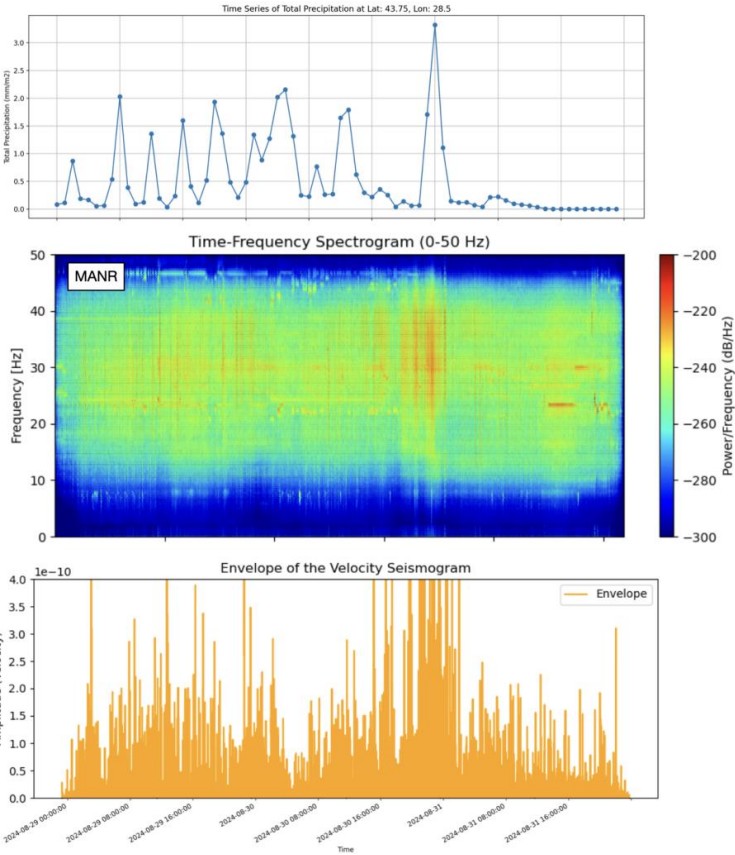

*Figure 4. High frequency (30-50Hz) observations of the storm at station MANR. a. Tiem series of total precipitation per hour from ERA5 at the grid location of station MANR. b. Spectrogram of the velocity time series for station MANR. c. Envelope of the velocity seismogram at station MANR.*

To visualise the signature of the storm passing over the network of broadband seismic stations in the coastal area, we also plotted the hourly precipitation values with the hourly root-mean-square amplitudes of the high-frequency (>30 Hz) seismic velocity envelopes recorded at seismic stations. Figure 5 shows three snapshots of hourly plots, illustrating a correlation between changing precipitation patterns from ERA5 data and the amplitudes of high-frequency seismic noise. This observation further supports the likelihood of a causal relationship.

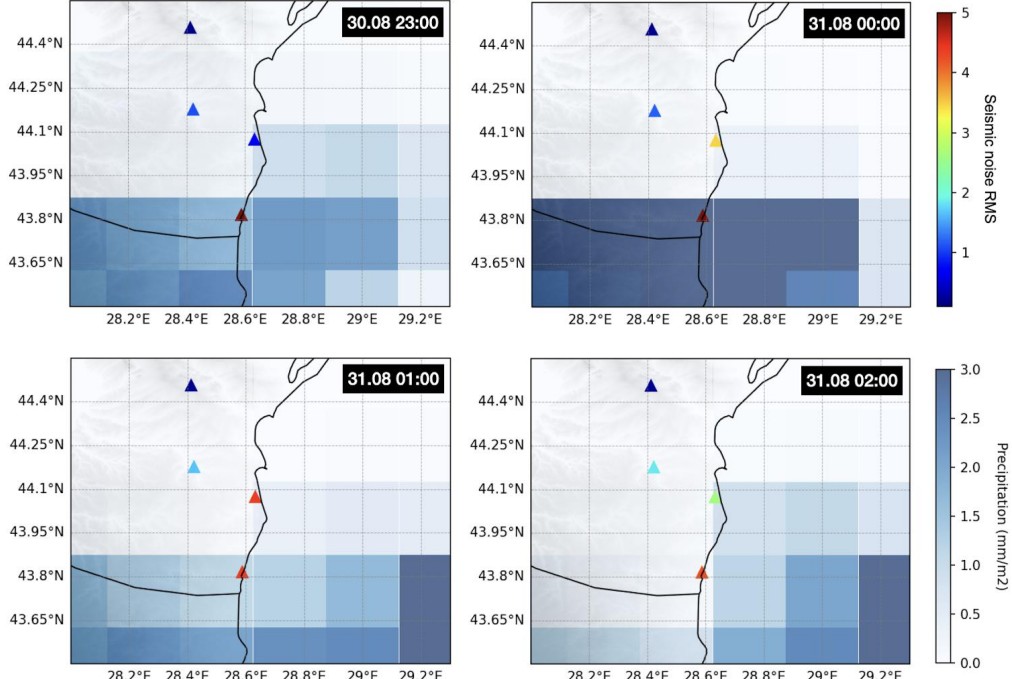

*Figure 5. Distribution of hourly RMS of high pass filtered (>30 Hz) seismograms and precipitation data.*
*Colours indicate hourly RMS amplitude of velocity envelopes filtered 30-50 Hz. Background coloured*
*grid indicates the total precipitation (mm/m2) from ERA5 data.*
The analysis of the microseismic noise frequency band is closely linked to the interaction between
ocean waves and the seafloor, which is influenced by storm conditions. To assess the storm's
impact, we analyze the PPSD (Probabilistic Power Spectral Density) of noise recorded at several
stations during both storm and quiet days, using the latter as a baseline. Figure 6 shows examples
of PPSD at stations MANR and MFTR (Figure 2), revealing differences in PSD amplitudes across
the primary and secondary microseismic bands. These differences indicate the presence of high-
intensity wind-driven waves and swell energy in the sea.
The secondary microseismic band shows a significant rise in amplitude during storms, driven by
wind-induced pressure fluctuations in the shoaling seafloor (Figure 3, Ebeling et al., 2012). On
quiet days, the PSD remains consistently lower, typically staying below the -120 dB threshold.
This stark contrast emphasizes the role of atmospheric conditions in modulating seismic noise,
with storms causing a notable increase in energy across both frequency bands. The temporal
evolution of the PSD values (Figure 7) further highlights the storm's impact, with fluctuations
corresponding to changes in environmental factors, reinforcing the connection between storm
activity and the observed seismic signals.



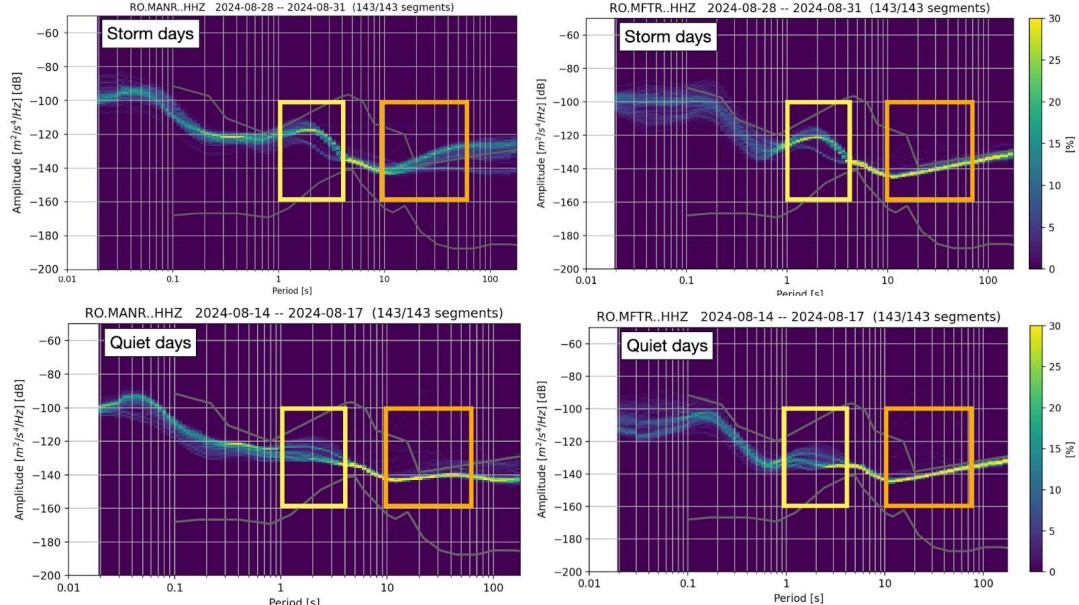


*Figure 6. Probabilistic Power Spectral Density (PPSD) plots for two seismic stations near the Black Sea
coast capturing the target storm signal in the microseismic bandwidths (marked with rectangles). The top
panels show the PPSD distributions across frequencies, indicating the probability of power spectral
density values in percentage for days including the Black Sea storm. The bottom panels show PPSD for
days with no recorded events.*



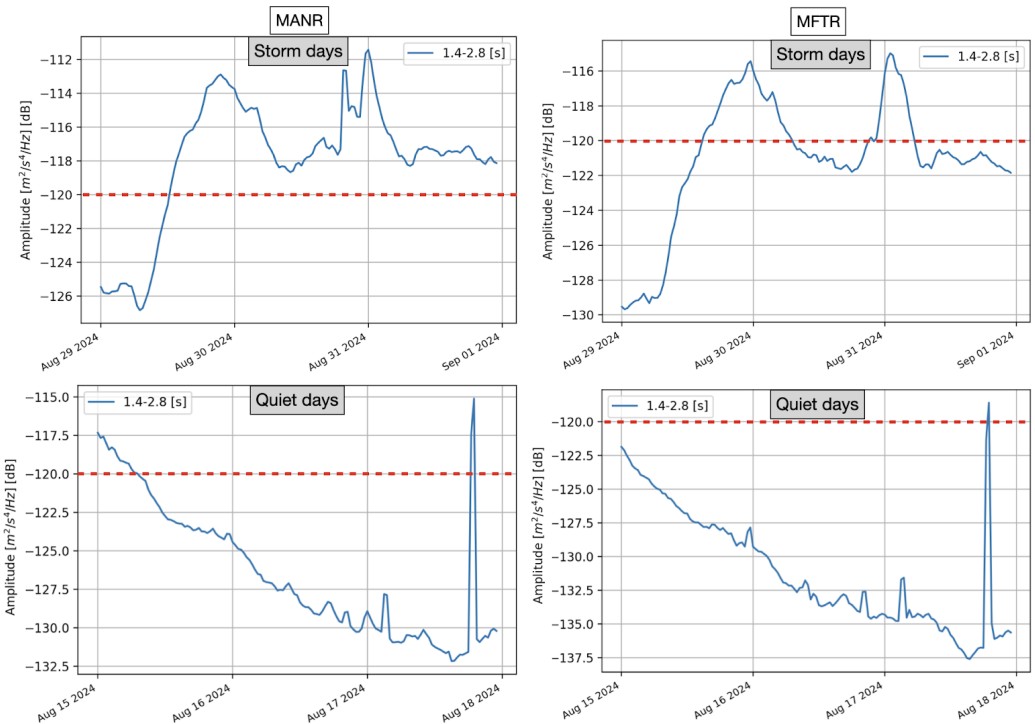

*Figure 7. Temporal PSDs for two seismic stations near the Black Sea Coast in the secondary microseismic band showing significantly higher values (>-120 dB) during stormy days compared to days without recorded precipitation.*

## 4.1 Infrasound and satellite lightning observations

### 4.1.1 Single station feature extraction

The evolution of time-frequency features over the duration of the Black Sea storm revealed distinct patterns in the infrasound signal (Figures 8 and 9). Most features, including spectral flux, spectral centroid, and spectral rolloff, exhibited a clear peak or a slightly bimodal peak corresponding to the storm's peak intensity, indicating a strong relationship between these features and the storm's dynamics. The peak in these features was most pronounced during the storm's active periods, reflecting rapid changes in the atmospheric conditions. However, the zero-crossings feature did not show a similarly distinct pattern, with its evolution being less evident in relation to the storm's phases. This suggests that most spectral features are sensitive to storm-related acoustic shifts.



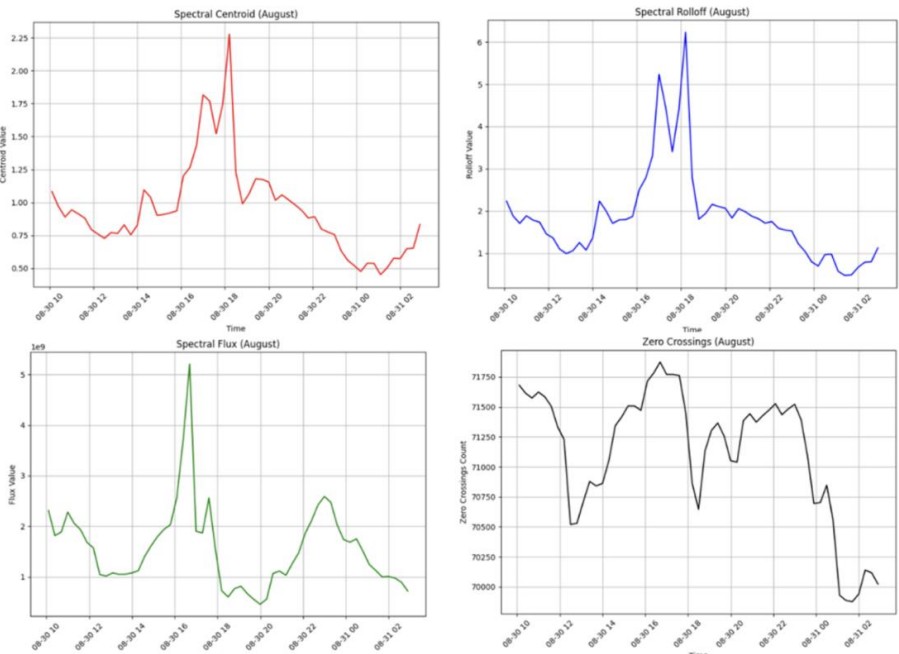

*Figure 8. Time-frequency feature analysis for the single-station infrasound signal recorded at AGIR during the Black Sea Storm.*

K-means clustering of the segmented infrasound signal, based on extracted time-frequency features, further revealed the relationship between the storm's acoustic dynamics and its temporal evolution, using 7 clusters for the enhancement of both detection of storm phases and event detection. The segmented signal highlighted distinct phases of the storm, with each segment corresponding to clusters representing specific changes in spectral content. These segments exhibited clear matches with the storm's progression, indicating that the clustering technique effectively tracked variations in storm intensity and the corresponding acoustic features. Interestingly, the spectral content of the infrasound signal showed similarities to seismic signal envelopes, particularly in the high frequency ranges (Figure 9), which may suggest a connection between the atmospheric pressure waves detected by infrasound and the ground vibrations captured by seismic instruments. This overlap implies that both seismic and infrasound signals could be complementary in capturing different aspects of storm dynamics, with seismic signals reflecting ground vibrations and infrasound capturing the atmospheric processes.

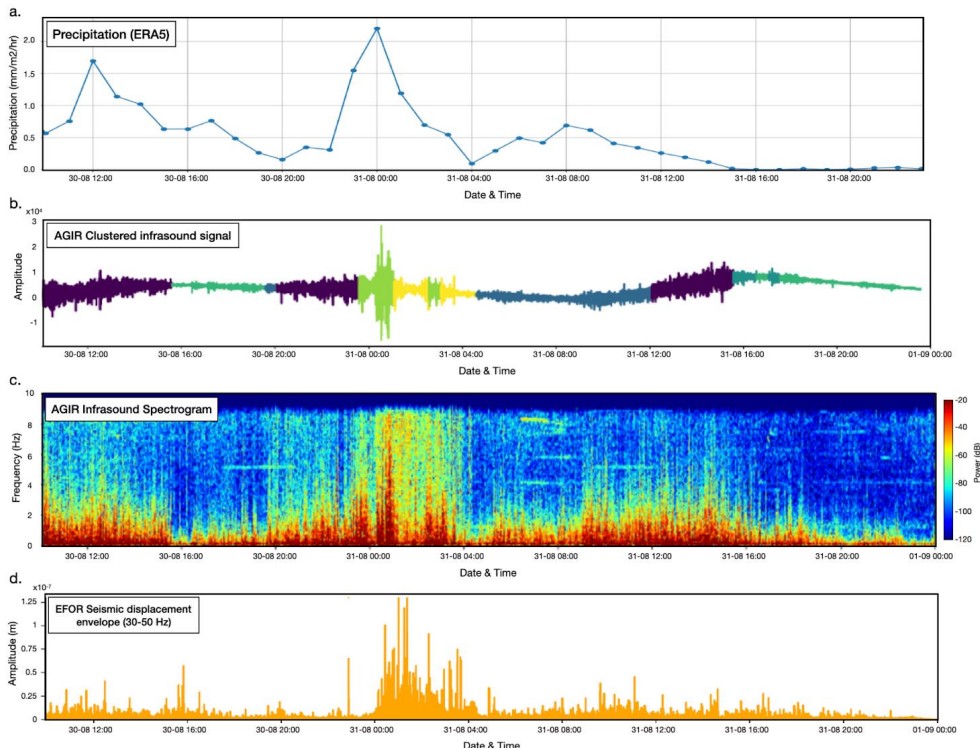

392

*Figure 9. Clustering results of infrasound signals recorded at AGIR from the August Black Sea storm event compared with precipitation data and seismic signal from EFOR station. a. Average precipitation data from 1ºx1º around AGIR. b. Raw infrasound signal recorded at the AGIR sensor during the period of August 30–31, 2024, with different segments color-coded according to the cluster they belong to, based on K-means clustering of 30-minute time-frequency feature windows. c. The corresponding spectrogram generated using Blackman windowing with 128 samples and 70% overlap; d. Seismic displacement envelope at station EFOR, filtered between 30-50 Hz.*

## 4.1.2 Array analysis and lightning detection

The PMCC algorithm allowed us to isolate coherent infrasound signals and estimate their propagation parameters, such as back-azimuth and arrival times, across the sensor network. These detections were cross-referenced with data from the MTG Lightning Imager, which recorded nearly 11,000 lightning strikes within a 50 km radius of the station during the Black Sea storm. The selected range ensured that the detected infrasound signals could be reliably linked to nearby lightning activity.

Approximately 1,100 infrasound detections were identified within the relevant frequency range of 0.6 to 7 Hz (Figure 10). These signals primarily consisted of long-duration wave trains, characterized by frequent amplitude peaks, likely associated with electrical discharges from lightning. The dominant frequency of these infrasound signals was approximately 3 Hz, with amplitudes varying between 0.01 and 3.4 Pa. These acoustic signatures, identified through the PMCC method, provide valuable insights into the storm's behavior, correlating infrasound signals



with specific lightning events detected by the MTG Lightning Imager and enhancing our
understanding of the atmospheric effects during the storm.

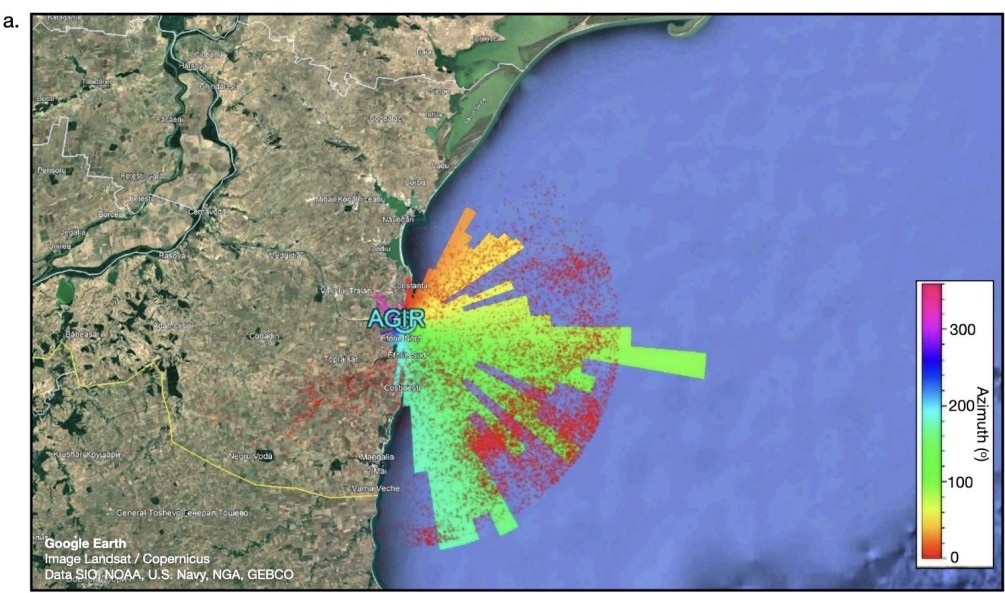

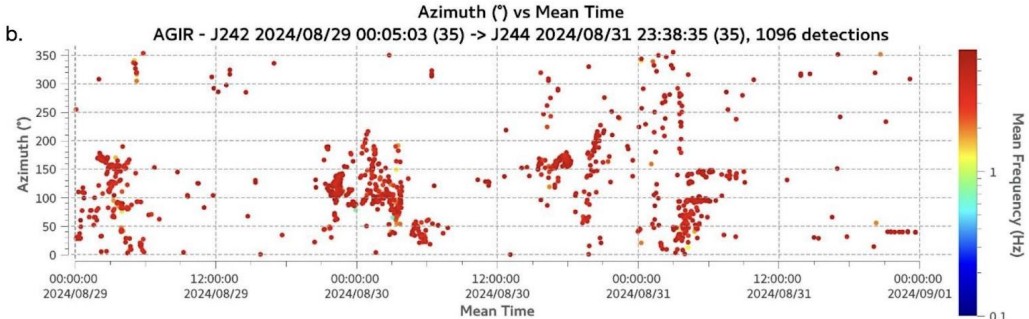

*Figure 10. a. Polar histogram of infrasound detections from the AGIR station, displayed with © Google*
*Earth, along with the locations of lightning strikes detected by the MTG Lightning Imager system for the*
*period from August 29 to 31, 2024. The geographical position of the AGIR infrasound station is also shown*
*on the map. b. diagram of high-frequency detections from the AGIR infrasound station between August 29*
*and 31, 2024.*
**4.4 GNSS-derived precipitable water vapor trends**
The analysis of daily GNSS-derived precipitable water vapor (Figure 11) reveals clear temporal
variations, with the highest PWV values consistently recorded on stormy days (>900 mm per day
on DOY 240–243). Notably, the peak values occurred between DOY 241 and DOY 243 (Figure
11b), when the heaviest rainfall was observed (Figure 1). Coastal stations showed extremely high
PWV values (>950 mm per day) compared to inland stations (>800 mm per day), with a slight
decrease in PWV as we moved inland (Figure 11a). This spatial distribution highlights the
geographical gradient of atmospheric moisture, with the highest PWV concentrations near coastal



areas, decreasing gradually toward the north. Interestingly, some inland stations (BUCU, PGNL,
RMSR) recorded their peak PWV on DOY 255, corresponding to the onset of the Boris storm,
another significant extreme rainfall event that swept through Central and Eastern Europe
(Athanase et al., 2024).
Elevated PWV was observed as early as DOY 239 (Figure 12a), suggesting that the tropospheric
moisture loading began to increase several days before the onset of the rainfall. This accelerated
increase in PWV can serve as an early indicator of a developing weather system. Remarkably,
although HAR1, located inland, did not directly experience the extreme rainfall, it exhibited
similar PWV behavior to coastal stations, suggesting that GNSS stations, even outside the
immediate storm zones, can capture atmospheric signals indicative of intense precipitation. This
finding offers a valuable precedent, showing that PWV measurements at GNSS stations not
directly in the storm's path can still provide critical insights into moisture dynamics at the
tropospheric level.

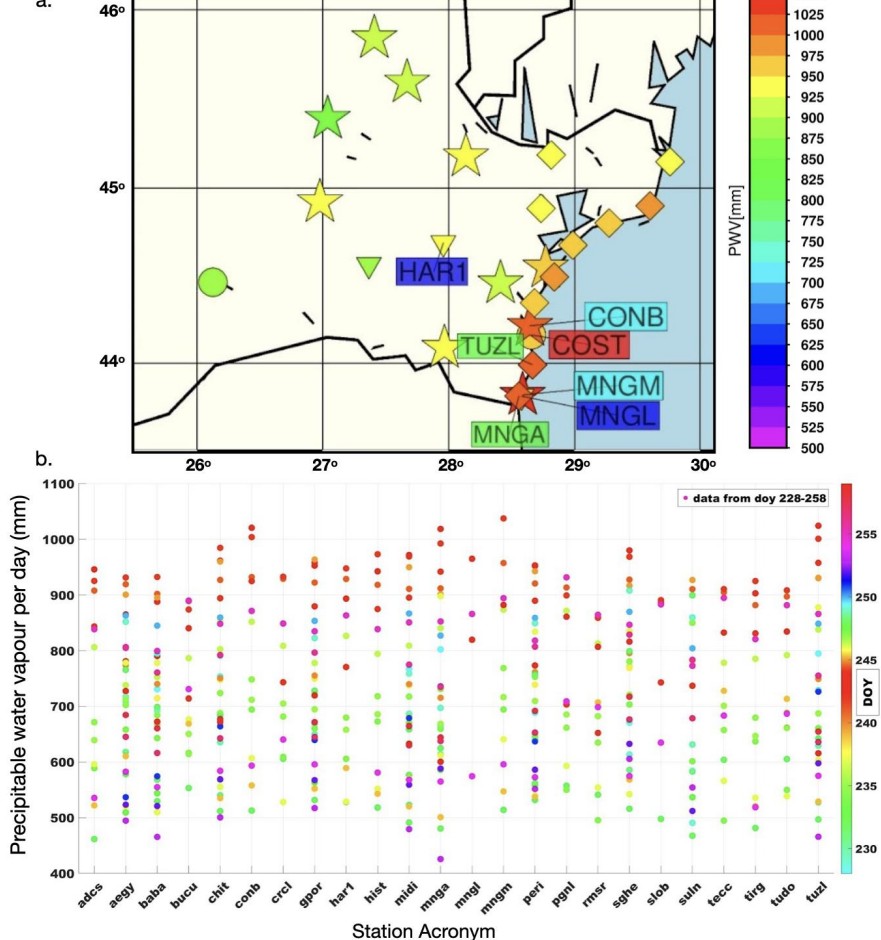






*Figure 11. a: Map of GNSS stations coloured as a function of PWV estimated a day before the heavy rainfall in August 2024. b: Daily PWV values for each station showing the difference between rainy days (red circles) and non-rainy days (coloured as a function of Julian day index).*

Shifting focus to hourly PWV data, Figure 12b shows the results from the MNGA station, which recorded the heaviest rainfall in the study area. Notably, MNGA also showed a rapid buildup of PWV, reaching values greater than 44 mm/hr just a few hours before the storm event. This rapid increase in PWV strongly suggests that the accumulation of atmospheric moisture is a precursor to extreme weather events, such as intense rainfall and storms. This observation aligns with known atmospheric dynamics, where a significant increase in water vapor content precedes heavy precipitation, providing further evidence of the potential for GNSS-based PWV monitoring to serve as an early warning tool for extreme weather.

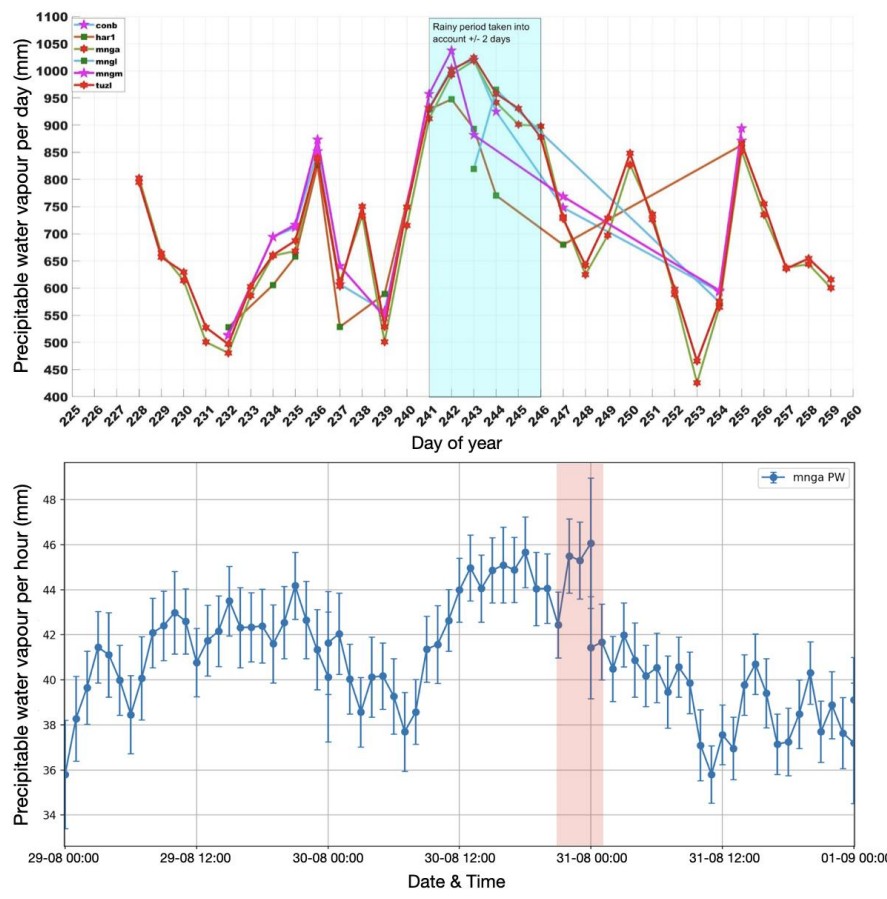

*Figure 12. Time series of precipitable water vapour per day estimated at several GNSS stations (Figure 2), shown over the period of a month centered on the storm event (highlighted in blue); b. Hourly rates of PWV for station MNGA with the heaviest recorded rainfall at the location of the station, highlighted in blue.*



## 5. Discussion

The integration of infrasound, seismic, and GNSS data in monitoring the extreme storm event over the Black Sea provides valuable insights into the dynamics of storm behavior and demonstrates the potential of non-conventional sensors for meteorological analysis. Infrasound data, for instance, revealed a clear acoustic signature of lightning activity, with signals detected in the range of 0.6 to 7 Hz corresponding to electrical discharges. The high frequency of infrasound detections (around 1,100) supports its utility as a reliable tool for tracking storm-related phenomena, particularly lightning, which is difficult to capture with traditional methods. However, the signals did not always perfectly align with lightning strikes, indicating that other factors, such as the movement of storm systems or variations in atmospheric conditions, may influence infrasound signatures. This suggests that refining the correlation between infrasound signals and lightning activity could be an avenue for future research, particularly in cases of sparse lightning or in remote regions.

A key aspect of the analysis was the use of unsupervised learning methods, such as K-means clustering, to identify patterns in the infrasound data. This approach segmented the infrasound signals into distinct clusters, suggesting the evolution of different phases of the storm. By identifying these phases, we can correlate shifts in spectral content with variations in storm intensity. However, a major goal for future work is to perform joint clustering with seismic data, which could provide a more comprehensive understanding of the storm's acoustic and seismic dynamics. Seismic data, particularly high-frequency seismic noise, closely resembled infrasound spectral decay, suggesting a coupling between the two types of signals. This spectral similarity may indicate that both infrasound and seismic signals are influenced by the same atmospheric and oceanic processes, such as pressure fluctuations caused by rainfall, wind, and storm-induced waves. The coupling between seismic and infrasound signals further emphasizes the need to integrate these data sources, as they capture different but complementary aspects of storm behavior. Future studies using joint clustering techniques will be crucial in enhancing the detection of storm phases and improving the understanding of the coupling between seismic and infrasound data.

Seismic data alone also showed a strong connection between high-frequency seismic noise and heavy rainfall, supporting previous studies that linked seismic signals to rainfall intensity. The distinction between high-frequency and low-frequency seismic noise is particularly noteworthy. High-frequency seismic noise correlated with precipitation, while low-frequency signals were associated with wave height and storm-driven winds. This suggests that different seismic frequencies capture distinct storm dynamics, with high-frequency signals reflecting localized rainfall impacts and low-frequency signals tied to broader atmospheric and oceanic interactions. This dual-frequency approach provides a more nuanced interpretation of seismic data in storm monitoring, highlighting its complexity.

The temporal variations observed in GNSS-derived integrated precipitable water vapor (PWV) provide valuable insights into atmospheric moisture dynamics before extreme weather events. The pronounced increase in PWV, particularly in the days leading up to and during the storm (DOY 241-243), supports the link between elevated atmospheric water vapor and precipitation. Notably, the buildup of PWV starting as early as DOY 239 suggests that rising moisture levels in the troposphere can serve as an early indicator of impending extreme rainfall. Even stations located up to 130 km inland, such as HAR1, recorded similar PWV trends, indicating that GNSS stations outside direct storm zones can still provide crucial atmospheric data. Hourly PWV trends further



revealed a rapid increase several hours before precipitation, with values exceeding 44 mm/hr,
highlighting the accumulation of moisture just before heavy rainfall. These findings align with the
notion that increasing atmospheric moisture acts as a precursor to intense precipitation,
highlighting the potential of GNSS-based PWV monitoring as a real-time tool for tracking
moisture and understanding short-term atmospheric fluctuations.
The integration of GNSS, infrasound, and seismic data provides a more comprehensive
understanding of storm dynamics than any single data source alone. The synergy between these
diverse sensor types allows for the detection of atmospheric moisture, lightning activity, rainfall-
induced seismic signals, and storm-driven oceanic interactions. Future research should focus on
refining unsupervised learning algorithms for infrasound and seismic signal classification,
optimizing joint clustering techniques, and improving the integration of these data sources to
enhance storm forecasting and early-warning systems. We believe this multi-sensor approach
holds promise for improving our ability to predict extreme weather events, understand their
impacts, and mitigate associated risks.

## 518    6. Conclusions

This study presents a comprehensive analysis of a record-breaking storm over the Black Sea, using
a combination of GNSS, infrasound, and seismic data to capture the dynamics of extreme weather
events. Our findings underscore the power of multi-sensor networks in enhancing the
understanding of storm behavior, particularly in the context of atmospheric moisture, lightning
activity, and storm-induced seismic signals. GNSS-derived integrated precipitable water vapor
(PWV) indicates a clear buildup of atmospheric moisture hours before the onset of heavy rainfall,
providing valuable insights into the lead-up to extreme precipitation events. Infrasound and
seismic data further complemented this analysis, with infrasound serving as a reliable tool for
tracking lightning activity and seismic data revealing the link between rainfall intensity and high-
frequency seismic noise.
The storm we analyzed is not only a significant meteorological event but also serves as a powerful
example of how climate change may be influencing the frequency and intensity of extreme weather
phenomena. Record-breaking storms like this are increasingly being recognized as evidence of
shifting atmospheric conditions, driven by global climate change. The integration of GNSS,
infrasound, and seismic data provides a more nuanced and holistic view of storm dynamics,
highlighting the need for advanced monitoring systems to predict and respond to such extreme
events. Moving forward, the combination of these innovative tools holds great potential for
improving early-warning systems, enhancing storm forecasting, and better understanding the
impacts of climate change on atmospheric and oceanic processes.

## 538    7. Code availability

Seismic data were processed with the open-source python framework for seismology Obspy
(Beyreuther et al., 2010). Infrasound data was processed with the WinPMCC software (Le Pichon
et al., 2010) developed by CEA/DASE (French Atomic Energy Commission, Environmental
Assessment and Monitoring Department) and open-source Python libraries for signal processing.
Some of the figures were made with GMT (Generic Mapping Tools, Wessel et al., 2019). The
GNSS data was processed using Gamit/Globk (Herring et al., 2020) developed by Massachusetts
Institute of Technology (http://www-gpsg.mit.edu/gg/).



## 8. Data availability

Seismic data are part of the Romanian National Seismic Network maintained by the National institute for Earth Physics (NIEP, www.infp.ro) and are freely available in the miniseed format via EIDA (European Integrated Data Archive, https://www.orfeus-eu.org/data/eida/). GNSS data are available for download from NIEP (http://gps.infp.ro/#/download) and are provided in the standardized RINEX v2 format, with 24-hour files sampled at 30-second intervals. Infrasound data at AGIR are available to download from NIEP via FDSN dataselect web service. Hourly hydro-meteorological data were obtained from the Copernicus Climate Change Service, Climate Data Store (https://doi.org/10.24381/cds.bd0915c6), ERA5 dataset (Hersbach et al., 2023). Lightning data came from Meteosat Third Generation Lightning Imager operated by EUMETSAT (The European Organisation for the Exploitation of Meteorological Satellites, https://www.eumetsat.int/).

## 9. Author contribution

**Laura Petrescu**: Conceptualization, Methodology, Software, Formal analysis, Data Curation, Writing-Original Draft, Visualization, Funding acquisition; **Bogdan Antonescu**: Conceptualization, Writing-Review & Editing, Visualization; **Sorin Nistor**: Software, Formal Analysis, Data curation, Visualisation, Writing-Review & Editing; **Iustin Floroiu**: Methodology, Software, Formal analysis, Data Curation, Writing-Original Draft, Visualization; **Dragoş Ene**: Software, Formal analysis, Data Curation, Writing-Review & Editing; **Daniela Ghica**: Software, Formal analysis, Data Curation; **Constantin Ionescu:** Funding Acquisition, Resources, Project administration; **Andrei Anghel:** Methodology, Supervision; **Mihai Datcu**: Methodology, Supervision, Funding Acquisition, Resources, Project administration.

## 10. Acknowledgments

We would like to thank the technicians and staff at NIEP for their support in installing, maintaining, and ensuring the proper functioning of the equipment used in this study. Additionally, we appreciate the efforts of those involved in data formatting and preparation (Cristian Neagoe, Eduard Nastase, Victorin Toader) which were essential for this work.

## 11. Financial support:

This work was carried out in the framework of the "Competence Center for Climate Change Digital Twin for Earth forecasts and societal redressment" Project PNRR- DTEClimate nr. 760008/31.12.2023, subproject Reactive "The Research center for climate change due to natural disasters and extreme weather events", supported by the Ministry of Research, Innovation and Digitalization of Romania.

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
