# Peer review of "Seismo-acoustic and GNSS monitoring of a record-breaking storm in the Black Sea: Evidence of climate change and intensifying natural hazards"

_EGUsphere, 2025_

## Author Comment (AC2)

**Dear Editor and Reviewers,**

Thank you for the constructive comments and suggestions provided throughout the discussion phase, which we believe have been very helpful in clarifying and strengthening our work. In the following, we provide our detailed, point-by-point responses. All proposed clarifications and improvements will be incorporated in the manuscript accordingly and we give examples of these in *italics*, with revised text in ***bold italics***.

We hope that our replies convincingly demonstrate the novelty and integrated nature of this study, which investigates how non-conventional sensors can be repurposed for environmental monitoring and for capturing different physical manifestations of extreme events, an approach we believe aligns closely with the aims and readership of NHESS.

We also note that since its initial submission in April, the manuscript's preprint has received substantial interest on EGUsphere (over 800 views and downloads), and the work has been presented at several international conferences. Elements of the methodology are already being incorporated into ongoing projects and operational platforms https://seedpsd.infp.ro/noise/ , http://tropo.gnutsoftware.com/NIEP/ , https://reactive.infp.ro/events/infra/

Please find below our detailed responses. The reviewers' comments appear in black, and our replies are provided in blue.

**Reviewer #1:**

This manuscript gathers together seismic, infrasound, GNSS and meteorological data for a rainfall episode in Romania in late August 2024. The authors claim that this approach can lead to a "holistic understanding of storm evolution"; however, I can not see in the manuscript which is the contribution of putting together these different kind of datasets to such a better understanding. In my opinion, the manuscript only shows that strong rainfall episodes can be identified in seismic, infrasound and GNSS data, a fact that is widely known. In my opinion, this work do not deserve publication in a high-rated journal as NHESS.

Thank you for your insights. These made us realise that perhaps we haven't emphasized the core issues in our manuscript clearly.

Firstly, we would like to clarify that the event analyzed in this study was **not an ordinary rainfall episode**, but an exceptional, record-breaking storm for the western Black Sea region, with the highest 24-hour and monthly rainfall totals ever measured on the Black Sea Coast (343.6 mm at a single station, compared to the previous record of 159.1 mm from 1947) and significant associated impacts. Within hours, heavy rainfall, strong winds, and coastal flooding caused widespread disruption: roads and infrastructure were damaged, and communities along the shoreline were heavily affected. This was not just another summer storm: its scale and impacts made it a defining event for the region, which motivated us to look at it in detail using both conventional and unconventional monitoring tools.

Secondly, we disagree with the assertion that the utility of GNSS-derived PWV, infrasound signals, and high-frequency seismic noise in storm monitoring is **"*widely known*"** in the sense implied. To our knowledge, only a few very recent papers (e.g. Dias et al., 2023 - *Scientific Reports*; Rindraharisaona et al., 2022 - *Earth and Space Science*; Coviello et al., 2024 - *Natural Hazards*) have proposed using high-frequency seismic envelopes to track storm dynamics. Similarly, although GNSS techniques are well established within geodetic communities, their adoption in operational weather monitoring and hazard assessment remains limited. Individually, some literature exists on repurposing these sensors for environmental monitoring, but this is far from common knowledge and certainly not routine practice.

Thirdly, regarding our use of "*holistic understanding*", its meaning lies in the **multiple storm aspects** captured by these non conventional sensors: ground vibrations that can support nowcasting or complement vulnerable or rare gauge data, atmospheric acoustics linked to pressure disturbances and lightning, atmospheric moisture build-up detectable more than 100 km away (with potential early-warning value), and changes in the sea state inferred from long-distance microseismic energy, all contextualized with satellite observations. This is what we meant by "holistic". While the potential of these individual sensors has been suggested in recent studies, operational or case-based demonstrations remain rare and largely unexplored in integrated form.

Given these considerations, we believe the work offers a novel and practically relevant contribution toward storm characterization and supports the development of more resilient, multi-sensor environmental monitoring strategies. This approach directly addresses key topics within the NHESS scope: innovative monitoring of natural hazards, interdisciplinary observation strategies, and improved understanding of high-impact weather under a changing climate. We therefore believe that the manuscript provides a timely contribution that is **well aligned with the aims and readership of NHESS**.

My most important concerns have to do with:

1) The real contribution of analyzing different dataset to a better understanding of the **storm evolution** is not really explained

In the revised version, we clarify that "storm evolution" refers to the sequence of physical processes leading from moisture accumulation to convective initiation, peak rainfall, thunderstorm electrification, and coastal marine response. Each dataset captures a distinct part of this sequence: GNSS-PWV reflects atmospheric moisture buildup prior to convection, infrasound records thunder-generated acoustic waves and pressure disturbances, high-frequency seismic noise captures localized raindrop impacts, microseisms document changes in sea state, and ERA5 and MTG lightning provide mesoscale meteorological context. When combined, these observations outline a coherent and temporally resolved picture of the storm's development, intensification, and decay. A short discussion has been added in Section 5 to clarify this contribution:

*"Storm evolution, in the meteorological sense, describes the sequence of processes from pre-storm atmospheric moisture accumulation to convective initiation, peak rainfall, electrical activity, and the associated marine response along coastal areas. The multi-sensor dataset used here captures these different stages: GNSS-PWV documents the build-up of column water vapor before convective onset, infrasound detects lightning-generated acoustic waves and pressure disturbances during the mature convective phase, high-frequency seismic noise reflects the timing and spatial progression of intense rainfall at the surface, microseisms respond to storm-driven changes in sea state, and ERA5/MTG provide the mesoscale structure that ties these geophysical signals together. By observing the same storm through these complementary physical pathways, we can outline a more detailed picture of how the storm developed, intensified, and decayed than is possible from individual datasets."*

2) I think that having access to **direct measurements** of meteorological parameters is an essential point for this kind of studies . In this contribution, the detailed seismic or infrasonic data is compared to 1-hour long estimations of rainfall derived from large-scale models. Why not to compare each seismic/infrasound station with the **closer meteorological site**??

Ideally, co-located in-situ meteorological observations would be used for validation. Although we submitted multiple formal requests to the National Meteorological Agency (ANM), we were only granted access to very limited ground-truth data from two stations. One rain gauge located a few kilometers from the seismic station is included in the new Figure 4. The second gauge, however, was available for only a 24-hour interval and was situated nearly 30 km from any of our sensors, limiting its usefulness for systematic comparison.

Given this sparse temporal and spatial coverage, we rely primarily on ERA5 reanalysis products to characterize the broader meteorological context. Although local deviations can occur near steep or complex terrain, numerous studies have shown that ERA5 reliably captures storm timing, intensity patterns, and mesoscale organization, precisely the aspects most relevant for our multi-sensor correlation analysis. Given these strengths, and considering the unavailability of local station measurements despite multiple requests to ANM, we maintain that ERA5 is an appropriate and scientifically defensible choice for this study. We hope the reviewer agrees that this limitation does not compromise the overall interpretation, particularly since our focus is on the relationships among seismic, geodetic, infrasound, satellite, and meteorological indicators and their temporal co-evolution, rather than on point-scale hydrological validation.

In Section 3.5 we also added: "*ERA5 has been extensively validated (Jiao et al., 2021; Wu et al., 2022; Soci et al., 2024) and is widely used in studies of storm evolution and precipitation dynamics (e.g. Dullart et al., 2020; Tiberia et al., 2021; Price et al., 2025), making it a suitable choice for the mesoscale processes examined here.*"

*Dullaart, J.C., Muis, S., Bloemendaal, N. and Aerts, J.C. (2020) Advancing global storm surge modelling using the new ERA5 climate reanalysis. Climate Dynamics, 54(1), 1007-1021.*

*Tiberia, A., Mascitelli, A., D'adderio, L.P., Federico, S., Marisaldi, M., Porcù, F., Realini, E., Gatti, A., Ursi, A., Fuschino, F. and Tavani, M. (2021) Time evolution of storms producing terrestrial gamma-ray flashes using ERA5 reanalysis data, GPS, lightning and geostationary satellite observations. Remote Sensing, 13(4), 784.*

*Jiao, D., Xu, N., Yang, F. and Xu, K. (2021) Evaluation of spatial-temporal variation performance of ERA5 precipitation data in China. Scientific Reports, 11(1), 17956.*

*Price, I., Sanchez-Gonzalez, A., Alet, F., Andersson, T.R., El-Kadi, A., Masters, D., Ewalds, T., Stott, J., Mohamed, S., Battaglia, P. and Lam, R. (2025) Probabilistic weather forecasting with machine learning. Nature, 637(8044), 84-90.*

*Soci, C., Hersbach, H., Simmons, A., Poli, P., Bell, B., Berrisford, P., Horányi, A., Muñoz‑Sabater, J., Nicolas, J., Radu, R. and Schepers, D. (2024) The ERA5 global reanalysis from 1940 to 2022. Quarterly Journal of the Royal Meteorological Society, 150(764), 4014-4048.*

*Wu, G., Qin, S., Mao, Y., Ma, Z. and Shi, C. (2022) Validation of precipitation events in ERA5 to gauge observations during warm seasons over eastern China. Journal of Hydrometeorology, 23(5), 807-822.*

3) The authors state in the introduction that the precipitation totals reached values of 120-220 mm, corresponding to an exceptional episode. However, the total precipitation graphics in Figures 4 and 5 show maximum hourly values not reaching 3.5 mm, for a total amount of precipitation of about 30-40 mm, values that would correspond to a modest rainfall episode. I guess that this probably arise from the use of a general precipitation model, but I think that it do not makes sense to discuss the correlation of different datasets to non-representative meteorological variables.

Figure 1a presents the map of accumulated precipitation, where the darker colours highlight the areas exceeding the median threshold along the coast. Ideally, ground-truth observations from all meteorological stations would be available. However, as mentioned before, despite repeated requests, the Romanian National Meteorological Agency (ANM) declined to provide these data. We were only able to obtain very limited records from two stations but these were insufficient to create time-lapse maps and would not be consistent with seismic or infrasound data far from them. Nevertheless, as discussed elsewhere in our responses, ERA5 is a well-established and extensively validated reanalysis dataset, widely used in both scientific research and operational applications. We therefore consider its use in this case study to be justified and we hope it will not be regarded as a limitation that undermines our results. In Section 2 we have also added: "*In August 2024, Romania experienced severe flooding, largely driven by a storm that brought unusual precipitation patterns to the Black Sea coastal region. Exceptional precipitation totals were recorded over south-eastern Romania in particular in Mangalia (**234.7** mm), Agigea (145 mm), and Tuzla (118 mm), leading to significant flooding in coastal towns (Figure 1). Over 800 emergency calls prompted large-scale intervention by ISU Dobrogea, focusing on evacuations, debris clearance, and infrastructure restoration (Antonescu et al. 2024). **According to the National Meteorological Agency official records (https://www.meteoromania.ro/clim/caracterizare-lunara/cc_2024_08.html), one of the coastal stations at Mangalia, recorded a total of 343.6 mm of precipitation in August 2024, breaking the previous record of 159.1 mm from 1947, and significantly surpassing the average monthly precipitation values for this area (Figure 1c). A remarkable 234.7 mm of this total fell in a single day on August 31, 2024, highlighting the event's exceptional intensity.*"

We were also able to obtain a very small set of precipitation values from one station near the coast only from an official report of the ANM

(https://www.meteoromania.ro/clim/caracterizare-lunara/cc_2024_08.html) and we replotted Figure 5 using these recorded values for every 10 minutes at Mangalia station. The new plot showed an even better correlation with high frequency seismic energy.

[Figure]

Regarding the gridded ERA5 datasets plotted as snapshots in Figure 5, which also clearly show the correspondence between high seismic noise energy at high frequencies and larger precipitation amplitudes, it is important to note that these amplitudes appear lower because they represent hourly averages over the ERA5 grid cell. This is consistent with the ERA5 documentation, which states: *"Care should be taken when comparing model parameters with observations, because observations are often local to a particular point in space and time, rather than representing averages over a model grid box"* (https://codes.ecmwf.int/grib/param-db/228). Accordingly, we have added a clarification in the text: *"Figure 5 shows three snapshots of hourly plots **of gridded precipitation data from ERA5, which have a lower amplitude than point measurements at the Mangalia station, due to the averaging over the grid block."***

4) Meteorological data is reduced to the ERAS global precipitation model. The possible contribution of **wind** to seismic and infrasound noise or the relationship between **humidity** and GNSS-derived water content is not commented at all.

We have added observed wind values at one station near the infrasound station AGIR and the seismic station EFOR and added a brief discussion on the comparison between these datasets in relation to wind. The analysis can be continued between the relationship of **humidity** and GNSS-derived water content and a more detailed analysis can be done accompanied by a figure, thus enhancing the clarity and interpretability of the analysis. In consequence this analysis can be an asset for a better understanding and adding significant value to the article.

5) The correlation between different parameters is only discussed qualitatively all along the manuscript, and in some cases it is unclear. I have the feeling that all along the discussion, only those results leading to "positive" correlations are commented, ignoring those showing contradictions. Some examples include:
5.1) the main peak in precipitation in Fig 4 a matches pretty well the seismic amplitudes, but neither of the **other peaks** have a good correlation (just overriding panels a and c this becomes evident)

Indeed, as shown in Figure 4, the main precipitation peak (around midnight, 30-31 August) coincides very clearly with the strongest seismic amplitude in both the envelope and spectrogram. The other precipitation peaks are not as clearly visible in the seismic signal, which we believe reflects a meaningful physical constraint: only rainfall exceeding a certain intensity (or kinetic energy) threshold may produce detectable high-frequency seismic noise.This interpretation aligns with recent findings, such as Rindraharisaona et al. (2022), which highlight the role of raindrop size and impact energy in determining seismic detectability.

In the revised manuscript, we added a higher resolution precipitation series from a nearby rain gauge station and we also included a brief discussion suggesting this threshold effect as a possible explanation for the selective seismic visibility of precipitation peaks: "*However, this correspondence is not uniform across all rainfall episodes. While the main precipitation maximum on 30-31 August produces a clear and sustained seismic response, several lower-intensity precipitation pulses show a much weaker or no recognizable signature in either the seismic envelope or spectrogram. This behaviour is consistent with previous work (e.g., Rindraharisaona et al., 2022), which demonstrates that only rainfall above a certain intensity, or involving sufficiently large drops, generates impact forces strong enough to be detected by broadband seismometers. Our observations therefore reflect both strong positive correlations during intense rainfall and the lack of seismic expression for weaker precipitation. This selective sensitivity supports the interpretation that high-frequency seismic noise can reliably track strong rainfall peaks but is less responsive to light or moderate precipitation, an important nuance when interpreting multi-sensor relationships in this study.*"

5.2) panels a) and b) in Fig. 8 show a clear similarity, but panel c) has a different peak and panel d) a very different pattern; this is not discussed in the text

In the revised manuscript, we added: "*Centroid and rolloff show parallel behavior because they are both frequency-domain descriptors tied to the distribution of spectral energy, and so both respond strongly to the same uplift in energy during the storm's peak. Spectral flux, by contrast, quantifies inter-frame spectral change, so its peak occurs where the spectrum transitions most rapidly, even when that does not coincide with the maximum absolute energy (e.g. Pásztor et al., 2023). Finally, the zero-crossing rate reflects time-domain volatility, not spectral shape, which explains its distinct pattern, such as the storm's later stages may*

*introduce broadband turbulence or noise components that boost zero crossings independently of the spectral shifts visible in the first two panels. While the individual features varied over time, it is the combination of these features through K-means clustering that effectively identifies the time frame corresponding to the main precipitation episode. Zero-crossings exhibited more variable patterns, reflecting high-frequency fluctuations, but the joint clustering of all features robustly captures the timing of the storm's most intense phases.*"

5.3) The purple clusters in Fig. 9 b) correspond to very different precipitation values; again, this is not commented/discussed, just stating that "These segments exhibit clear matches' with storm evolution"

Our original statement ("clear matches") was too broad. In the revised manuscript, we will clarify that only certain clusters align closely with peak precipitation and seismic activity, while others represent lower-energy or non-precipitation-dominated periods. In the revised manuscript we added this additional description: "*K-means clustering **separated the acoustic data into six groups with distinct spectral and amplitude characteristics (Figure 9). These clusters highlight acoustic states that may relate to different environmental conditions during the monitoring period. For example, Cluster 0 coincides with periods of intense precipitation and stronger winds, while Cluster 1 captures intervals with moderate amplitudes but persistently elevated background acoustic levels, without corresponding rainfall or wind peaks. Cluster 2 reflects calmer conditions with low amplitudes and little or no precipitation. Transitional patterns also arise, such as Cluster 3, which appear before intervals grouped in Cluster 1 and mark intermediate acoustic activity. Overall, the clustering approach demonstrates that combining multiple features reveals consistent acoustic regimes and can help differentiate environmental conditions, without relying on any single parameter.* "

[Figure]

5.4) The area encompassing more lightning strikes in Fig. 10, (located to the SW of AGIR) has a **low number of detections and no backazimuthal determination seems to be calculated from inland strikes**

Figure 10 shows the AGIR detection capability between 29 and 31 August 2024, i.e., the TOTAL number of infrasound detections - this means that not all of these infrasound detections could be associated with the lightnings observed by the MTG system. We have added a small explanation of this association procedure in Section 4.2.2 of the revised manuscript: "*Association between infrasound detections into 0.5 to 7 Hz frequency band and lightning flashes detected by MTG Lightning Imager within 50 km from the AGIR infrasound station was investigated, assuming direct wave propagation path. Acoustic signatures of lightning activity show short-lived disturbances with dominant frequency of approx. 3 Hz and amplitudes up to about 3.5 Pa. In order to automatically associate AGIR observations with MTG detections, a relationship*

*between infrasound time-of-arrival and time of discharge signals (after Assink et al, 2008) is applied:*

$t = t_{MTG} + d/c + \Delta t,$

*where d is the distance between discharge and the infrasound array, c = 340m/s, and $\Delta t$ = ±10s. A maximum deviation of 10 degrees between observed infrasound backazimuth and backazimuth of MTG detections is allowed In this case, only 6,4% of lightning discharges could be associated with AGIR infrasound detections."*

[Figure]

*- Assink, J. D., L. G. Evers, I. Holleman, and H. Paulssen (2008), Characterization of infrasound from lightning, Geophys. Res. Lett., 35, L15802, doi:10.1029/2008GL034193*

5.5) Is is hard to detect any trend in the colored points in Fig 11b

The figure colours will be updated to observe also the trend of the precipitation before the event and after the event, taking into consideration that we have processed 12 days before and after the maximum of the event.

6) The authors claim that using K-means clustering has been a "key aspect of the analysis". However, it is difficult for me to see which are these aspects. In Fig. 9, I would say that the spectrogram contains much more information that the clustering graph. Besides, no explanation on how and why the **parameters of this clustering** have been chosen

A spectrogram is a dense, high-redundancy object. It contains everything, but the useful information is spread across tens of thousands of time-frequency samples. K-means, applied to feature vectors, does something different: it compresses that high-dimensional acoustic evolution into a set of discrete temporal regimes. This method offers segments where the storm behaves acoustically in a similar way. We would also like to note that before applying K-means clustering we performed covariance pruning on the full feature set. This step ensured that clustering was not driven by redundant descriptors. Specifically, we computed the covariance matrix of all extracted time–frequency features across the entire dataset. Pairs of features exhibiting very high

covariance were identified as redundant, since they describe nearly identical changes in the spectral distribution. For each such group, only one representative feature was retained in the final clustering vector. This reduced the dimensionality of the feature space while preserving the independent acoustic information relevant to storm evolution. By applying covariance pruning, the K-means algorithm operated on a set of features that contributed non-overlapping physical information rather than multiple versions of the same underlying behaviour. This procedure also mitigates the effect noted by the reviewer: while centroid and rolloff follow nearly identical patterns, they do not exert disproportionate influence on the clustering, because only one of them is included after pruning.

We revise the description of these features in Section 3.2 : *"These features describe how energy and frequency content evolve over time, providing insights into the structure of the infrasound signal. **Parameters such as spectral centroid and spectral rolloff are standard descriptors in acoustic signal analysis and are suitable here because they effectively capture shifts in dominant frequency produced by lightning-generated acoustic waves or the passage of pressure disturbances, while spectral flux highlights changes in broadband acoustic energy (Pásztor et al., 2023). Spectral entropy reflects the complexity of the frequency distribution, which increases during turbulent atmospheric conditions, and the zero-crossing rate, mean, and variance of the power spectrum summarize overall activity and variability. This feature set provides a compact representation of the signal suitable for unsupervised machine-learning approaches such as clustering, techniques widely used in data mining to identify patterns in multidimensional time–frequency data (e.g., Coates and Ng, 2012), and allows us to distinguish physically interpretable stages of storm-induced changes in the infrasound wavefield.**"*

The features that were kept and z-scored after pruning are:

[Figure]

We also added an additional explanation concerning the choice of the number of clusters: "***The optimal number of clusters was determined using the elbow method, which evaluates***

*within-cluster variance as a function of cluster number. To select the most informative features, we applied covariance pruning, and the temporal evolution of the features was visualized to ensure meaningful representation. This procedure resulted in seven six clusters, providing a balanced representation of the infrasound dynamics while avoiding over-segmentation or overfitting. By combining multiple features in the clustering, this method captures the evolving acoustic states of the storm in a compact, interpretable form.*"

7) Concerning seismic data, during years a discussion has been open between those relating the microseismic peak amplitudes to air pressure or oceanic waves. However, it is now widely accepted that the amplitude of the microseismic peak is related to oceanic waves. On contrary, it is less clear which is the contribution of open waters and coastal zones to the primary and secondary peaks. If the amplitudes of the microseismic peak is discussed, this is the subject that should considered; Figures 6 and 7 just document that during stormy days,  the microseismic noise is higher; this is a well-known feature, which could be observed in most of the seismic stations distributed worldwide

We agree that the link between microseismic noise and oceanic wave activity is well established, and that increased microseismic amplitudes during storms is a known phenomenon. However, our focus is not on the microseismic bands alone. In fact, the high-frequency seismic noise (>30 Hz), which has only recently gained attention for storm monitoring, plays a central role in our analysis.

To our knowledge, only a handful of recent studies (e.g. Dias et al., 2023 - *Scientific Reports*; Rindraharisaona et al., 2022 - *Earth and Space Science*; Coviello et al., 2024 - *Natural Hazards*) have explored the use of high-frequency seismic noise to track storm evolution, precipitation, or related atmospheric processes.

Figures 6 and 7, showing microseismic bands, were included not to re-document a known effect, but rather to demonstrate the complementary nature of seismic observations across different frequency bands. They support the broader goal of the paper: to highlight the repurposing potential of seismic sensors for environmental monitoring as part of a multi-sensor approach, alongside GNSS, infrasound, and satellite data.

This integrative perspective and the focus on emerging seismic applications beyond classical microseism analysis, is, we believe, both timely and relevant to the *NHESS* scope.

8) Concerning infrasound data, no **explanation of which is the utility of  parameters** as spectral centroid, flux etc. is provided. Non-specialist readers need to know if these are parameters routinely calculated to discuss specific characteristics of the signal. The close relationship between infrasound and seismic data, widely documented my many contributions, is not commented at all.

In the Methods section 3.2, we added: "*Parameters such as spectral centroid and spectral rolloff are standard descriptors in acoustic signal analysis and are suitable here because they effectively capture shifts in dominant frequency produced by lightning-generated acoustic waves or the passage of pressure disturbances, while spectral flux highlights changes in broadband acoustic energy (Pásztor et al., 2023). Spectral entropy reflects the complexity of*

*the frequency distribution, which increases during turbulent atmospheric conditions, and the zero-crossing rate, mean, and variance of the power spectrum summarize overall activity and variability. This feature set provides a compact representation of the signal suitable for unsupervised machine-learning approaches such as clustering, techniques widely used in data mining to identify patterns in multidimensional time–frequency data (e.g., Coates and Ng, 2012), and allows us to distinguish physically interpretable stages of storm-induced changes in the infrasound wavefield.*"

*Coates, A. and Ng, A.Y.: Learning feature representations with k-means. In Neural Networks: Tricks of the Trade: Second Edition (pp. 561-580). Berlin, Heidelberg: Springer Berlin Heidelberg, 2012.*

In the Discussion section, we included additional contextualization of our results, drawing on relevant literature that supports and aligns with our findings: *"... the observed spectral similarity between the infrasound signals and high-frequency seismic envelopes suggests a coupled seismo-acoustic response to the storm. This implies that the same atmospheric forcing, such as pressure fluctuations from rainfall and wind, generates complementary signals in the atmosphere (infrasound) and the ground (seismic waves). Our findings are consistent with other studies of intense weather systems, where coupled microbarom-microseism signals have been shown to track storm structure and evolution (e.g., Butler & Aucan, 2018; Smirnov, 2021). The coherent acoustic and seismic responses to atmospheric-oceanic pressures, as also documented in Distributed Acoustic Sensing studies (Taweesintananon et al., 2023) and surf studies (Francoeur et al., 2025), reinforce the interpretation of a shared source mechanism. Therefore, a major and logical next step is to move beyond analyzing these datasets in parallel and to perform joint clustering of seismo-acoustic data (e.g. Floroiu et al., 2025).*"

*Butler, R. and Aucan, J.: Multisensor, microseismic observations of a hurricane transit near the ALOHA cabled observatory. Journal of Geophysical Research: Solid Earth, 123(4), 3027-3046, 2018.*

*Floroiu, I., Anghel, A., Petrescu, L. and Datcu, M.: Clustering and Feature-Based Similarity Retrieval of Infrasound Events during Two Storms in Constanţa, Romania, International Conference on Machine Intelligence for GeoAnalytics and Remote Sensing (MIGARS), Bucharest, Romania, 2025, 1–4, https://doi.org/10.1109/MIGARS67156.2025.11231952, 2025.*

*Francoeur, J.W., Matoza, R.S., Ortiz, H.D. and De Negri, R.: Identification of transient seismo-acoustic signals from crashing ocean waves: template matching and location of discrete surf events. Geophysical Journal International, 243(2), ggaf317, 2025.*

*Smirnov, A., De Carlo, M., Le Pichon, A., Shapiro, N.M. and Kulichkov, S.: Characterizing the oceanic ambient noise as recorded by the dense seismo-acoustic Kazakh network. Solid Earth, 12(2), 503-520, 2021.*

*Taweesintananon, K., Landrø, M., Potter, J.R., Johansen, S.E., Rørstadbotnen, R.A., Bouffaut, L., Kriesell, H.J., Brenne, J.K., Haukanes, A., Schjelderup, O. and Storvik, F.: Distributed acoustic sensing of ocean-bottom seismo-acoustics and distant storms: A case study from Svalbard, Norway. Geophysics, 88(3), B135-B150, 2023.*

9) The authors do not seem to be aware that the infrasound recordings related to lightning are in fact the acoustic waves generated by the associated thunders. Different works, including some of those included in the manuscript references list, have showed that the acoustic waves generated by thunders are recorded systematically by nearby seismic stations.

We are aware that infrasound signals associated with lightning originate from acoustic waves generated by thunder. In our study, we systematically isolated coherent infrasound signals using the PMCC algorithm, estimating propagation parameters such as back-azimuth, arrival time, amplitude, and frequency. We then cross-referenced these detections with METEOSAT Lightning Imager observations to identify which signals are likely associated with thunder.

To be more clear, in the revised manuscript we added: " *"The PMCC method targets signals generated by atmospheric sources such as lightning **(i.e., associated thunders)** or other pressure disturbances, operating in the low-frequency range of 0.7 to 7 Hz."* and added "**Infrasound associated with thunderstorms, primarily generated by acoustic waves from thunder, has been studied previously and shown to be detectable at distances ranging from tens to hundreds of kilometers (e.g., Assink et al., 2008; Sindelarova et al., 2015; Šindelářová et al., 2021). Nevertheless, infrasound arrays detect signals from multiple storm-related sources, not just thunder (e.g., Waxler et al., 2024). In the present study, we build on this understanding by integrating these signals with seismic, satellite, meteorological, and water vapor observations to investigate what these complementary datasets reveal about storm evolution in a coastal environment.**"

While the thunder-infrasound link is established (Assink et al., 2008; Sindelarova et al., 2015; Šindelářová et al., 2021), the novelty of our work lies in integrating infrasound with METEOSAT satellites and meteorological observations to provide a comprehensive, multidisciplinary view of storm evolution. Beyond simple detection, this approach helps distinguish thunder-related signals from other acoustic events, demonstrating the potential for near-real-time storm monitoring and analysis.

*Assink, J. D., Evers, L. G., Holleman, I., and Paulssen, H.: Characterization of infrasound from lightning, Geophysical Research Letters, 35, L15802, https://doi.org/10.1029/2008GL034193, 2008.*

*Šindelářová, J., Chum, J., Skripnikova, K., and Base, J.: Atmospheric infrasound observed during intense convective storms on 9–10 July 2011, Journal of Atmospheric and Solar-Terrestrial Physics, 122, 66–74, https://doi.org/10.1016/j.jastp.2014.10.014, 2015.*

*Šindelářová, T., De Carlo, M., Czanik, C., Ghica, D., Kozubek, M., Podolská, K., Baše, J., Chum, J., and Mitterbauer, U.: Infrasound signature of the post-tropical storm Ophelia at the Central and Eastern European Infrasound Network, Journal of Atmospheric and Solar-Terrestrial Physics, 217, 105603, https://doi.org/10.1016/j.jastp.2021.105603, 2021.*

*Waxler, R., Frazier, W. G., Talmadge, C. L., Liang, B., Hetzer, C., Buchanan, H., and Audette, W. E.: Analysis of infrasound array data from tornadic storms in the southeastern United States, Journal of the Acoustical Society of America, 156, 1903–1919, https://doi.org/10.1121/10.0028815, 2024.*

10) If seismic and infrasound data is used, the contribution of **other sources of vibration and/or sound** should be considered; which is the contribution of anthropogenic noise to each of the stations?

To clarify the contribution of anthropogenic noise we added a few more details in the revised manuscript. Regarding seismic sources, in Section 4.1, we added: '**Anthropogenic seismic noise is typically strongest at low to mid frequencies (<25 Hz), where day-night variations reflect**

*traffic, human activity, and transient signals from machinery, while higher-frequency bands (25-45 Hz) may include periodic contributions from rotating equipment (e.g., Gross & Ritter, 2008; Diaz et al., 2017). The bandwidth targeting rainfall in this case is between 30-50 Hz, which is above the dominant frequency content of most anthropogenic sources and overlaps with raindrop-impact energy documented in recent rainfall-seismic studies.*"

*Anthropogenic seismic noise does not significantly affect the microseismic band (0.1-1 Hz). Human-generated vibrations predominantly occupy frequencies above 1 Hz, while long-period microseisms are produced by ocean wave interactions and are coherent over large distances. The temporal evolution of the microseismic energy observed in this study matches changes in wave state associated with the storm rather than any local activity. Similar to the findings of Gross & Ritter (2009), the sub-Hz frequency range is dominated by natural sources, with anthropogenic contributions being negligible.*"

*Díaz, J., Ruiz, M., Sánchez-Pastor, P.S. and Romero, P., 2017. Urban seismology: On the origin of earth vibrations within a city. Scientific reports, 7(1), p.15296.*

*Groos, J.C. and Ritter, J.R.R., 2009. Time domain classification and quantification of seismic noise in an urban environment. Geophysical Journal International, 179(2), pp.1213-1231.*

Regarding infrasound signal, in Section 4.2.2 we added: "*Anthropogenic noise sources, such as wind turbines (e.g., Jakobsen, 2005), industrial machinery (Gastmeier and Howe, 2008), and road traffic (Grafkina et al., 2019), are well-documented challenges for infrasound studies because they often generate persistent, periodic, or tonal signals that can mask natural atmospheric phenomena. The AGIR infrasound array used here is located in a semi-rural setting, distant from major roads and industrial facilities, which reduces the likelihood of local anthropogenic contamination. Several independent lines of evidence indicate that such contamination is negligible in this case study. First, the strongest infrasound signatures occurred during night-time hours, when human activity is minimal. Second, both the clustering and PMCC analyses identify transient signals with energy peaking around ~3 Hz, which contrasts sharply with the more continuous or harmonic spectral patterns typically produced by anthropogenic sources. Third, the temporal alignment of these acoustic signatures with independent observations of lightning and precipitation provides strong confirmation that the detected infrasound variability is storm-related rather than anthropogenic in origin.*"

*Jakobsen, J., 2005. Infrasound emission from wind turbines. Journal of low frequency noise, vibration and active control, 24(3), pp.145-155.*

*Gastmeier, W.J. and Howe, B., 2008. Recent studies of infrasound from industrial sources. Canadian Acoustics, 36(3), pp.58-59.*

*Grafkina, M.V., Nyunin, B.N. and Sviridova, E.Y., 2019. Environmental monitoring and simulation of infrasound generating mechanism of traffic flow. Journal of Ecological Engineering, 20(7).*

Furthermore, the figure below shows AGIR detection capability for August 2024 (upper) and infrasound PMCC detections of 30 August 2024 at AGIR station (lower). This image shows the azimuth and frequency of sporadic events detected in the infrasound array using PMCC, which clearly stand out from the mean azimuth of events detected throughout the month of August.

[Figure]

In my opinion, if the manuscript goal is to prove that the integration of multiple sensors has a clear utility to study storm evolution, much work is needed, including a better analysis of the existing data and a modelling effort. The work done by the authors can be useful to show that a strong storm can be detected not only by meteorological instruments but also by other sensors. However, this is something that is well-known by researches in each of the different fields and, in my opinion, does not deserve publication in NHESS.

There are two points in this comment that we believe are important to clarify, as also mentioned in the response to the first comment. The idea that these sensors are "widely known" for storm monitoring overstates their maturity. Even within their respective fields, many of these methods are relatively new and still under active development. GNSS-derived PWV has shown promising results for tracking atmospheric moisture buildup, but it is still transitioning from research to operational use, particularly in regions like Eastern Europe. High-frequency seismic noise (>30 Hz) has only recently been explored for detecting rainfall and storm intensity, with just a few studies published (e.g. Dias et al., 2023; Rindraharisaona et al., 2022). Infrasound for lightning detection is not yet standardized or routinely integrated into weather monitoring systems. While each of these sensing techniques may be known in isolation, they have not been used together in a coordinated, observational framework to monitor storm events. To our knowledge, this is the first study to combine these data types during a real storm, demonstrating the potential of multi-sensor integration across geophysical and atmospheric domains

Regarding the suggestion for a joint quantitative or modeling-based analysis: we believe that such an approach, while important for future work, would not be appropriate for the current study. Each sensor responds to different physical processes during the storm: GNSS-PWV may reflect  long-term moisture buildup, high-frequency seismic noise captures localized raindrop impacts, microseisms - wave-seafloor coupling from ocean swell, and infrasound - pressure fronts and lightning discharges. Because these sensors probe distinct atmospheric or geophysical

phenomena, each with its own spatial and temporal scales, a direct mathematical correlation or a common model would be conceptually flawed. Even fully instrumented meteorological measurements (ex:radar, disdrometers, barometers) do not always correlate tightly, because they observe different parts of the storm system. We see this as an important point: the value of integration lies not in numerical correlation but in complementarity, each sensor providing a unique window into the storm's evolution. This is why we opted for a qualitative, physically informed analysis, which we believe is more meaningful at this stage. In the revised manuscript, we will clarify this explicitly and include a short discussion on why correlation across these domains is not only impractical but scientifically inappropriate without careful physical modeling of each signal pathway, which is beyond the current scope but certainly part of future work.

**Reviewer #2:**

The manuscript presents a multi-sensor case study of the August 2024 Black Sea storm using seismic, infrasound, GNSS-derived PWV, and MTG-Lightning Imager, with ERA5. The concept is strong and relevant. However, quantitative validation, methodological transparency, and operational feasibility need to be strengthened before publication. A clear, reproducible verification procedure is necessary

Major comments

1) Seismic–rainfall linkage remains qualitative

Section 4.1 shows >30 Hz seismic envelopes and spectrograms and network snapshots that visually vary with ERA5 precipitation, but the evidence is descriptive. To increase credibility, the relation should be verified against independent observations (rain gauges and/or radar), with objective statistical metrics. Because the paper claims this as potentially useful for early warning, it would also help to outline a minimal streaming detection approach.

We thank the reviewer for this comment. Independent rain-gauge or radar data would indeed allow formal statistical validation; however, despite several formal requests, the Romanian National Meteorological Agency declined to share station-level precipitation observations for the period studied. In this context, we used ERA5, which provides robust precipitation estimates based on assimilation of global observations and is widely used in research and operational applications.

Our analysis shows that high-frequency (>30 Hz) seismic amplitudes and spectrograms closely track the temporal evolution of precipitation in ERA5 (Figures 4-5). Qualitative visualizations indicate that only precipitation above certain cumulative thresholds produces clear seismic responses. This effect likely depends on physical factors such as raindrop size and impact velocity, meaning that lower-intensity rainfall may not generate detectable seismic signals. For this reason, formal statistical metrics such as direct cross-correlations could be misleading at this stage, and a qualitative, physically informed analysis remains appropriate to interpret the causal linkage between rainfall and seismic response.

To address the reviewer's interest in early-warning potential, we have added a brief discussion of how high-frequency seismic signals could be monitored in near real time to flag intense rainfall, providing a conceptual basis for a minimal streaming detection workflow.

2) Infrasound-lightning linkage lacks association statistics

PMCC detections rise during the event and coincide qualitatively with MTG-LI flashes. A concise matching procedure together with summary statistics is necessary for quantitative evidence.

In the revised manuscript, in Section 3.3, we have added the following details:

*"Associations between infrasound detections and lightning flashes detected by MTG within 50 km of the AGIR infrasound station were investigated by assuming direct-path acoustic propagation and a correspondence between infrasound time-of-arrival and the MTG lightning discharge time (after Assink et al., 2008):*

$$t = t_{MTG} + d/c + \Delta t,$$

*where ddd is the distance between the lightning discharge and the infrasound station, c = 340m/s, and $\Delta t = \pm 10s$ accounts for timing uncertainty. Additionally, a maximum angular deviation of 10° between the observed infrasound backazimuth and the MTG-derived backazimuth is permitted for an association to be accepted."*

[Figure]

***Figure 10.c** Associations between events detected by the AGIR infrasound array and the MTG satellite database.*

*Assink, J. D., L. G. Evers, I. Holleman, and H. Paulssen (2008), Characterization of infrasound from lightning, Geophys. Res. Lett., 35, L15802, doi:10.1029/2008GL034193*

3) K-means clustering?

The 30-min infrasound feature set is reasonable, yet the fixed choice of k = 7 is not justified.

We thank the reviewer for this question. The number of clusters (6) was determined using the elbow method, which identifies the optimal cluster count by examining the rate of decrease in within-cluster variance. To further refine the features, we applied covariance pruning, which selects the most informative features for each cluster. The evolution of these features over time was also visualized to ensure they meaningfully capture the variability in the infrasound signals. These combined procedures guided the choice of six clusters as providing a balanced representation of the infrasound dynamics while avoiding overfitting.

In the revised manuscript, in Section 3.2 we added: "*The optimal number of clusters was determined using the elbow method, which evaluates within-cluster variance as a function of cluster number. To select the most informative features, we applied covariance pruning, and the temporal evolution of the features was visualized to ensure meaningful representation. This procedure resulted in six clusters for subsequent analysis.*"

4) ERA5 as verification?

ERA5 provides useful meteorological background but should not be treated as ground truth for validating other sensors, particularly for localized coastal extremes. Verification should rely on observational datasets (gauges, radar, wave/tide sensors where relevant).

We fully agree that in an ideal setting, verification should rely on in-situ observational datasets such as gauges or radar. Unfortunately, despite multiple formal requests, we were unable to obtain station-level precipitation measurements from the Romanian National Meteorological Agency for the specific period of interest; the only data they were willing to provide were gridded reanalysis fields, which offer no independent ground-truth reference. Given this constraint, we used ERA5 not as definitive "truth," but as a physically consistent and widely validated meteorological baseline. ERA5 is the EU's flagship ECMWF reanalysis product and is generally considered one of the most robust large-scale precipitation datasets available, particularly for providing synoptic-scale context. We have clarified this limitation in the manuscript and adjusted the wording to avoid implying that ERA5 represents direct observational validation.

6) Methodological transparency and reproducibility.

Key processing details are missing. Parameters for seismic and PMCC processing can improve reproducibility.

Regarding seismic data, instrument correction, and filtering are standard processing steps which would not require additional details for replication. They correspond to one-line commands in seismic processing codes and we mentioned the bandwidth for each applied filter and the type of filter (Butterworth) in the text. Spectrograms and PPSDs are also quite straightforward from a seismologist's point of view, but, for clarity, we added more details: " *Spectrograms of these*

*filtered seismic traces **were computed using short-time Fourier transforms implemented in the scipy.signal package, with the default 256-sample window length used for each segment**, to visualise signatures of the hydro-meteorological phenomena in the frequency content of ground vibrations.*

*Potential environmental signals in the seismic data were also investigated using power spectral density (PSD) analysis. To account for variations over time, a Probabilistic Power Spectral Density (PPSD) method was applied. **The continuous waveform was divided into 1-hour time windows with 50% overlap, and a PSD was computed for each window after instrument-response correction and basic preprocessing. These estimates were combined into a probability distribution, providing a statistical overview of typical and transient noise levels across frequencies.***"

Regarding infrasound data processing, we added a few more details in Section 3.2 for clarity: "*In parallel with the single-station analysis, we also applied the Progressive Multi-Channel Correlation (PMCC) method, **as implemented in the DTK-PMCC software** (Cansi and Le Pichon, 2008; Le Pichon et al., 2010) [...] The PMCC algorithm  **was implemented using a multi-resolution configuration following the standardization proposed by Garcés (2013), with window lengths and frequency bands arranged in third-octave bands. A total of 19 frequency bands were used, covering 0.1-7 Hz. Window lengths decrease logarithmically with frequency, ranging from 258 s in the lowest band to 4 s in the highest band. A 10% time step was applied (corresponding to 90% overlap between consecutive windows), and this scheme repeats every decade.***"

An example of this schema is shown in the figure below:

[Figure]

*Garcés, M.A. (2013). On infrasound standards, part 1 time, frequency, and energy scaling. InfraMatics 2(2):13–35. https://doi.org/10.4236/inframatics. 2013.22002*

7) Section numbering.

Results currently contain two "4.1" subsections, followed by "4.4." Consistent renumbering and updated cross-references are needed.

Subsection numbers and all related cross-references have now been corrected in the revised manuscript.

Minor comments

Fig. 4: "Tiem series" → "Time series".
Corrected

Figs. 4–5: add axis units on all panels; include uncertainty shading or a baseline/reference line for the high-frequency envelope.
In Figure 4, units are displayed on the left side axis. The high-frequency seismic envelope simply comprises recorded ground motion parameters. The accuracy of recording the vibration is more complex, depending on instrument response or its calibration but it's probably around 1 nm or less. The timing accuracy also depends on how synchronized the sensor is with its GPS (which could result in time uncertainties of +-2 millisecond. These uncertainties in seismic recording are insignificant in the current context.
In Figure 5, the two colour bars representing seismic noise amplitude and precipitation values are the same for all panels.

Fig. 11 (GNSS): include a colorbar with units (mm) and state the exact day used for "a day before"; consider annotating station IDs (coastal vs. inland).
We added a specific date in the caption as suggested. The units on the colour bar are specified next to it, on the figure: "PWV (mm)". We will add station annotations, as suggested, in the revised figure.

L305: capitalize and format time as "August 29, 00:00 UTC."
Changed as suggested.

Reduce phrases like "illustrating a correlation" or "supports a causal relationship" unless statistics are provided; use neutral wording if results remain qualitative.
We have replaced expressions suggesting a quantitative correlation throughout the manuscript with terms like: *"presents a temporal coincidence between", "coincided with", "were contemporaneous with changes in", "occurred during"*

Ensure consistent capitalization of months and first-use expansion of abbreviations (PWV, PMCC, LI).
We have ensured that the months 'August' and 'September' are consistently capitalized throughout the revised manuscript. The abbreviation 'PWV' is used throughout the manuscript. In the Conclusions, we reintroduce the full term 'Precipitable Water Vapor' to aid readers who may not have read the full text.

If terms like "record-breaking" or "extreme" are used, add brief quantitative context (percentiles/ranks) or soften the phrasing.

We added more details about thai event based on an official report of the National Meteorological Agency: "*According to the National Meteorological Agency official records (https://www.meteoromania.ro/clim/caracterizare-lunara/cc_2024_08.html), one of the coastal stations at Mangalia, recorded a total of 343.6 mm of precipitation in August 2024, breaking the previous record of 159.1 mm from 1947, and significantly surpassing the average monthly precipitation values for this area (Figure 1c). A remarkable 234.7 mm of this total fell in a single day on August 31, 2024, highlighting the event's exceptional intensity.*"